# Blocking mitochondrial pyruvate import in brown adipocytes induces energy wasting via lipid cycling

Michaela Veliova[1,2] iD, Caroline M Ferreira[2,3], Ilan Y Benador[2,4], Anthony E Jones[1] iD, Kiana Mahdaviani[4], Alexandra J Brownstein[2,5], Brandon R Desousa[1], Rebeca Acín-Pérez[2] iD, Anton Petcherski[2] iD, Essam A Assali[2,6] iD, Linsey Stiles[2], Ajit S Divakaruni[1], Marc Prentki[7], Barbara E Corkey[4], Marc Liesa[1,2,*] iD, Marcus F Oliveira[3,**] iD & Orian S Shirihai[1,2,4,***] iD

## Abstract

Combined fatty acid esterification and lipolysis, termed lipid cycling, is an ATP-consuming process that contributes to energy expenditure. Therefore, interventions that stimulate energy expenditure through lipid cycling are of great interest. Here we find that pharmacological and genetic inhibition of the mitochondrial pyruvate carrier (MPC) in brown adipocytes activates lipid cycling and energy expenditure, even in the absence of adrenergic stimulation. We show that the resulting increase in ATP demand elevates mitochondrial respiration coupled to ATP synthesis and fueled by lipid oxidation. We identify that glutamine consumption and the Malate-Aspartate Shuttle are required for the increase in Energy Expenditure induced by MPC inhibition in Brown Adipocytes (MAShEEBA). We thus demonstrate that energy expenditure through enhanced lipid cycling can be activated in brown adipocytes by decreasing mitochondrial pyruvate availability. We present a new mechanism to increase energy expenditure and fat oxidation in brown adipocytes, which does not require adrenergic stimulation of mitochondrial uncoupling.

**Keywords** futile cycle; malate aspartate shuttle; metabolism; mitochondrial pyruvate carrier; thermogenesis
**Subject Category** Metabolism

## Introduction

Brown adipose tissue (BAT) is the main site of non-shivering thermogenesis (Cannon, 2004; van Marken Lichtenbelt *et al*, 2009; Virtanen *et al*, 2009; Cypess *et al*, 2009). Fat oxidation is activated in BAT when the sympathetic nervous system releases norepinephrine after sensing cold temperatures, which activates adrenergic receptors on BAT to increase thermogenesis. This activation causes a stimulation of lipolysis, mediated by protein kinase A actions on lipolytic proteins, ultimately activating mitochondrial uncoupling protein 1 (UCP1) by free fatty acids (Cannon, 2004; Divakaruni & Brand, 2011). UCP1 is exclusively expressed in thermogenic adipocytes and dissipates the energy of mitochondrial proton gradient to generate heat, instead of conserving it as ATP (Cannon, 2004). Numerous studies demonstrated the beneficial effects of BAT activation in the treatment of metabolic disorders (Poekes *et al*, 2015). Although recent publications have proposed mechanisms to increase energy expenditure in BAT independent of hormonal stimulation (Kazak *et al*, 2015; Ikeda *et al*, 2017; Mahdaviani *et al*, 2017; Deng *et al*, 2018), currently there is only one known pharmacological approach to increase BAT energy expenditure in humans, which is the use of β3-adrenergic receptor agonists, such as mirabegron (O'Mara *et al*, 2020). While promising, potential undesirable side effects may arise from systemic β3-adrenergic stimulation.

Increased glucose uptake is a hallmark of BAT activation, as shown by [18]fluorodeoxyglucose positron emission tomography–computed tomography ([18]F-PET-CT) images in humans and mice (Cypess *et al*, 2009; van Marken Lichtenbelt *et al*, 2009; Virtanen *et al*, 2009; Hankir *et al*, 2017). Pyruvate, the end-product of glycolysis, has multiple metabolic fates in the cytosol and in the mitochondria. The first step of mitochondrial pyruvate oxidation requires pyruvate import across the inner mitochondrial membrane, which is mediated by the mitochondrial pyruvate carrier (MPC) (Herzig *et al*, 2012; Bricker *et al*, 2012). MPC is a protein complex that transports pyruvate in a proton motive force-dependent manner (Papa *et al*, 1971; Herzig *et al*, 2012). Several studies have shown that MPC activity determines the fuel oxidized by mitochondria, with direct consequences to cell function and fate (Vacanti *et al*, 2014; Yang *et al*, 2014; Vanderperre *et al*, 2015; Vanderperre *et al*,

1   Department of Molecular and Medical Pharmacology, David Geffen School of Medicine at UCLA, Los Angeles, CA, USA
2   Division of Endocrinology, Department of Medicine, David Geffen School of Medicine at UCLA, Los Angeles, CA, USA
3   Instituto de Bioquímica Médica Leopoldo de Meis, Universidade Federal do Rio de Janeiro, Rio de Janeiro, RJ, Brazil
4   Nutrition and Metabolism, Graduate Medical Sciences, Boston University School of Medicine, Boston, MA, USA
5   Molecular Cellular Integrative Physiology, University of California, Los Angeles, CA, USA
6   Department of Clinical Biochemistry, School of Medicine, Ben Gurion University of The Negev, Beer-Sheva, Israel
7   Department of Nutrition, Université de Montréal, Montreal Diabetes Research Center and  CRCHUM, Montréal, Canada
    *Corresponding author. Tel: +1 310 206 7319; E-mail: mliesa@mednet.ucla.edu
    **Corresponding author. Tel: +55 2122708647; E-mail: maroli@bioqmed.ufrj.br
    ***Corresponding author. Tel: +1 310 825 5160; E-mail: oshirihai@mednet.ucla.edu

2016; Divakaruni *et al*, 2017). Once in the matrix, mitochondrial pyruvate undergoes simultaneous decarboxylation to acetyl-CoA and condensation with oxaloacetate to produce citrate. Citrate can then be used as a precursor of malonyl-CoA, the first intermediate of *de novo* fatty acid synthesis. At the same time, malonyl-CoA also inhibits the rate-controlling step of mitochondrial fatty acid oxidation by blocking carnitine palmitoyl transferase 1 (CPT1). The role of mitochondrial pyruvate oxidation driven by MPC activity in brown adipocytes remains elusive, as the assumptions are that i) pyruvate might not be required as a fuel in BAT mitochondria, as brown adipocytes mostly oxidize fatty acids when mitochondrial uncoupling is adrenergically activated to generate heat and ii) the preferred fate of pyruvate from increased glycolysis in BAT would be lactate, to allow glycolysis to compensate for uncoupling and provide the ATP not produced by uncoupled mitochondria. To gain insight into the role of mitochondrial pyruvate transport and metabolism in BAT energy metabolism, we determined the effects of blocking the MPC on energy expenditure. Here, we show that pharmacological and genetic inhibition of the MPC activates coupled fat oxidation in the absence of adrenergic stimulation, thus demonstrating that MPC inhibition increases ATP demand and fat expenditure. The increased flux in coupled fatty acid oxidation induced by MPC blockage is supported by glutamine consumption and the malate-aspartate shuttle (MASh), a previously understudied metabolic pathway in BAT. Interestingly, we show that acute MPC blockage has an additive effect promoting energy expenditure in adrenergically stimulated brown adipocytes, suggesting that MPC inhibition enhances fatty acid oxidation for ATP synthesis in mitochondria in which UCP1 is not activated. Finally, we demonstrate that the ATP-consuming glycerolipid/free fatty acid (GL/FFA) cycling (Reshef *et al*, 2003; Prentki & Madiraju, 2008) is responsible for increased ATP demand in response to MPC inhibition. Thus, we show that MPC activity limits energy expenditure in resting and activated brown adipocytes, rather than facilitating it. We conclude that MPC activity regulates BAT energy metabolism and its inhibition may represent a novel target to increase fat oxidation and energy expenditure independent of adrenergic stimuli, with potential benefits for treatment of metabolic diseases.

## Results

### Inhibition of the mitochondrial pyruvate carrier increases energy expenditure fueled by fatty acid oxidation

To determine the role of the mitochondrial pyruvate import in BAT fuel preference, we sought to assess the effects of MPC inhibition on fatty acid utilization in non-stimulated brown adipocytes. Mitochondrial pyruvate import was blocked using UK5099, a pharmacological inhibitor that covalently binds the MPC (Halestrap, 1975). To confirm that UK5099 at the concentrations used inhibited pyruvate oxidation by blocking pyruvate import and not by causing mitochondrial dysfunction, we compared the acute effect of UK5099 on mitochondrial respiration fueled by pyruvate versus succinate. To directly test mitochondrial pyruvate and succinate oxidation, cells were permeabilized to allow direct provision of pyruvate and succinate to mitochondria. Permeabilization prevented any metabolic rewiring induced by MPC blockage in intact cells to preserve

mitochondrial ATP synthesis by using other available fuels. UK5099 decreased oxygen consumption rates (OCR) in a dose-dependent manner in permeabilized brown adipocytes under pyruvate and malate (Fig EV1A), but not under succinate and rotenone (Fig EV1B). These data are in agreement with previous studies showing a specific effect of UK5099 on pyruvate oxidation at low micromolar concentrations (Divakaruni *et al*, 2013, 2017).

Previous studies demonstrated that upon MPC inhibition, fat mobilization and oxidation increased (Vacanti *et al*, 2014; Yang *et al*, 2014; Vanderperre *et al*, 2015; Vanderperre *et al*, 2016; Divakaruni *et al*, 2017). To test the role of the MPC in the fate of fatty acids in brown adipocytes, we used the fluorescent fatty acid analog Bodipy C12 558/568 (Bodipy C12), which can be oxidized and/or incorporated into lipid droplets (LDs) as triacylglycerides (TAGs). Primary brown adipocytes were incubated overnight with Bodipy C12 to stain lipid droplets and chase lipid droplet size and fluorescence after 120 min of UK5099 treatment. Lipid droplets were imaged using high resolution airyscan confocal microscopy in living cells (Fig 1A). We found that treatment with UK5099 significantly reduced lipid droplet content (Fig 1A and B). Reduced lipid droplet content by MPC inhibition was similar to the reduction induced by norepinephrine (NE) in brown adipocytes (Fig 1B), strongly suggesting that UK5099 increased lipolysis. Next, we tested whether MPC inhibition increased utilization of extracellular fatty acids by measuring the fate of exogenous fluorescently-labeled fatty acid using thin-layer chromatography (TLC). We pulsed brown adipocytes with Bodipy C12 for 24 h in the presence of either 100 nM UK5099, 1 μM NE or vehicle. Lipids were extracted from both media and cells and resolved by TLC (Fig 1C) (Rambold *et al*, 2015; Benador *et al*, 2018). Treatment with UK5099 increased Bodipy C12 uptake from the media similarly to NE, indicating that fatty acid uptake and utilization were increased upon MPC inhibition (Fig 1C and D). To measure C12 incorporation into TAGs, we quantified the amount of TAGs containing exogenously added C12 per amount of C12 uptake. We found that both UK5099 and NE induced a reduction in C12 incorporation into TAGs (Fig 1D). Altogether, these data provide evidence that MPC inhibition promotes fatty acid uptake by brown adipocytes, with the majority of these fatty acids going toward oxidation rather than storage as TAGs.

Based on the results in Fig 1A–D, we reasoned that MPC inhibition might affect mitochondrial respiration fueled by fatty acids. Thus, we measured the effects of MPC inhibition by UK5099 on mitochondrial OCR in non-stimulated brown adipocytes. We observed that inhibition of the MPC resulted in a dose-dependent increase in mitochondrial OCR of intact brown adipocytes (Fig 1E). The maximal increase in OCR by UK5099 was observed at 10 μM. This observation raised two questions: (i) What are the fuels driving the increase in OCR and (ii) what are the ATP-requiring processes that are activated upon MPC inhibition? We first wanted to examine what substrates were being oxidized to fuel the elevation in respiratory rates. Given that both lipolysis and fatty acid uptake were increased under UK5099 treatment (Fig 1A–D), we tested the requirement for fatty acids in UK5099-induced increase in OCR. We hypothesized that, if increased mitochondrial respiration induced by UK5099 is dependent on a rise in intracellular levels of fatty acids, then their sequestration would prevent the boost in respiratory rates upon MPC inhibition. To assess the role of increased free fatty acids availability in UK5099-induced OCR, we supplemented the

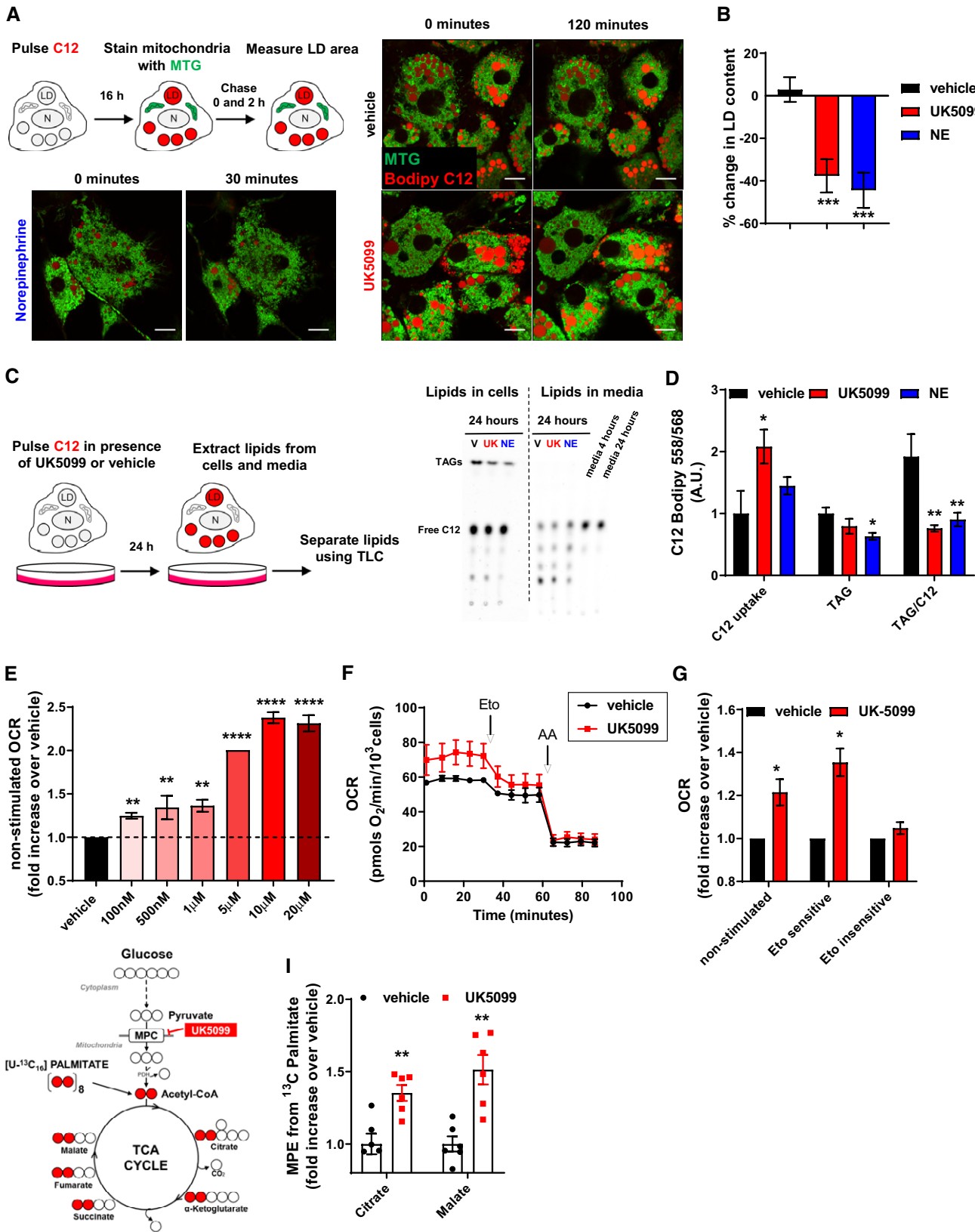

**Figure 1.**

Figure 1.  **Inhibition of the mitochondrial pyruvate carrier increases energy expenditure fueled by fatty acid oxidation.**

A, B   Effect of MPC inhibition on cellular lipid droplet content. (A) Live-cell super-resolution confocal imaging of primary brown adipocytes pre-stained overnight with the fatty acid tracer BODIPY C12 558/568 (Bodipy C12, red). Cells were stained with mitotracker green (MTG, green) prior to imaging. Cells were imaged before and 120 min after treatment with either vehicle (DMSO) or 100 nM UK5099. As a positive control, cells were imaged 30 min after treatment with 1 μM norepinephrine. LD, lipid droplet; N, nucleus. Scale bar = 10 μm. (B) Quantification of changes in lipid droplet cross-sectional area with indicated treatments from images shown in (A). Data are represented as percentage change in LD area at time = 120 min compared to time = 0 min. Data represent $n$ = 23–34 cells from 3 individual experiments. ***$P$ < 0.0001 by ANOVA, relative to vehicle.

C, D   Effect of MPC inhibition on fatty acid uptake and incorporation into triacylglycerides (TAGs). (C) Representative thin-layer chromatography (TLC) plate of lipids extracted from primary brown adipocytes and cell culture media. Cells were incubated with Bodipy C12 in presence of vehicle (V), 100 nM UK5099 (UK) or 1 μM norepinephrine (NE) for 24 h and triacylglyceride (TAG) synthesis (Lipids in cells) and fatty acid uptake (Lipids in media) were detected. The relative polarity of the lipid species determines the motility, with nonpolar TAG migrating the furthest. (D) Quantification of C12 uptake from media, TAG in cells and TAG per amount of free fatty acid uptake shown in (C), $n$ = 5 individual experiments. Note that UK5099 treatment decreases the amount of synthesized TAG per free fatty acid uptake, similarly to NE. *$P$ < 0.05, **$P$ < 0.01 by ANOVA, relative to vehicle.

E–G   Effect of UK5099 treatment on cellular energy expenditure. Fully differentiated primary brown adipocytes were pre-treated with vehicle (DMSO) or UK5099 at indicated concentrations for 2 h. Oxygen consumption rates (OCR) were measured in respirometry media supplemented with 5 mM glucose and 3 mM glutamine in the presence of vehicle or UK-5099. (E) Quantification of non-stimulated OCR from $n$ = 3–7 individual experiments. Data were normalized to vehicle for each individual experiment. Note that UK5099 increases OCR in non-stimulated brown adipocytes. **$P$ < 0.01, ****$P$ < 0.0001 by ANOVA, relative to vehicle. (F) Brown adipocytes were treated with 100 nM UK5099 or vehicle. Etomoxir (Eto; 40 μM) and antimycin A (AA; 4 μM) were injected where indicated. Representative OCR traces averaging 6 technical replicates. (G) Quantification of non-stimulated OCR, etomoxir-sensitive, and etomoxir-insensitive OCR from $n$ = 4 individual experiments. Data were normalized to vehicle for each individual experiment. *$P$ < 0.05 compared to vehicle by Student's $t$-test.

H, I    Effect of MPC inhibition on adipocyte fuel preference toward fatty acid. (H) Schematic representation of metabolite tracing using [U-$^{13}$C$_{16}$] palmitate. (I) [U-$^{13}$C$_{16}$] palmitate tracing in fully differentiated primary brown adipocytes treated with 5 μM UK5099 or vehicle for 24 h. Data show mole percent enrichment (MPE) of isotope-labeled substrate into respective metabolite. Data were normalized to vehicle for each individual replicate. $n$ = 6 technical replicates from 2 individual experiments. **$P$ < 0.01 compared to vehicle by Student's $t$-test.

Data information: All data are presented as mean ± SEM.

respirometry media with 0.1% fatty acid-free bovine serum albumin (BSA), which sequesters the excess intracellular free fatty acids (Alsabeeh *et al*, 2018). BSA prevented the stimulatory effects of UK5099 on mitochondrial respiration, suggesting that increased fatty acids availability supports respiration when MPC is inhibited (Fig EV1C).

To further test whether the increase in mitochondrial respiration induced by UK5099 is fueled by fatty acids, we used etomoxir to block a rate-controlling step of mitochondrial fatty acid oxidation catalyzed by carnitine palmitoyl transferase 1 (CPT1) (Kiorpes *et al*, 1984). The contribution of fatty acid oxidation to UK5099-induced respiration was assessed by determining the portion of OCR that is sensitive to etomoxir. Etomoxir was used at 40 μM, a concentration that specifically blocks CPT1 and has only minimal effects on other mitochondrial processes (Fig EV1D). Remarkably, the component of OCR that was sensitive to etomoxir was significantly higher in UK5099-treated brown adipocytes, strengthening our assertion that MPC inhibition promotes fatty acid oxidation (Fig 1F and G). On the other hand, the etomoxir-insensitive component of OCR was not affected by MPC inhibition (Fig 1G). These data suggest that MPC inhibition is primarily inducing oxidation of fat as compared to other fuels in brown adipocytes.

The above observations indicate that increased respiratory rates meditated by MPC inhibition were prevented if fatty acid import to mitochondria was blocked. However, blocking fatty acid utilization with etomoxir could be upregulating the utilization of other fuels to preserve mitochondrial ATP synthesis, thus underestimating the rates of fatty acid oxidation under MPC inhibition. Therefore, to unequivocally determine the real preference toward fatty acid oxidation when MPC is inhibited, we measured the incorporation of palmitate-derived carbons into tricarboxylic acid (TCA) cycle intermediates using gas-chromatography/mass spectrometry (GC/MS) (Fig 1H). [U-$^{13}$C$_{16}$] Palmitate and 5 μM UK5099 or vehicle were concurrently provided to non-stimulated brown adipocytes for 24 h.

Enrichment of $^{13}$C in TCA cycle metabolites was quantified in polar metabolite extracts from total lysates (Cordes & Metallo, 2019). Figure 1I shows that UK5099 increased the incorporation of $^{13}$C from labeled palmitate into citrate and malate, indicating that MPC inhibition increases fatty acid oxidation to acetyl-CoA that enters the TCA cycle (Fig 1I). Collectively, our data indicate that in the absence of NE stimulation, inhibition of the MPC increases energy expenditure fueled by fatty acid oxidation in mitochondria.

## MPC1 inhibition stimulates mitochondrial respiration coupled to ATP synthesis in brown adipocytes

Sympathetic stimulation of brown adipocytes increases mitochondrial fatty acid oxidation by stimulating lipolysis and uncoupling mitochondria. Consequently, acute treatment of brown adipocytes with norepinephrine (NE) increases mitochondrial fat oxidation to produce heat by activating UCP1. Thus, we next aimed to determine whether MPC inhibition would further increase fatty acid oxidation in activated brown adipocytes or whether it would be inert, as expected with MPC1 inhibition in uncoupled and depolarized mitochondria. We find that MPC inhibition by UK5099 further increased OCR in NE-stimulated brown adipocytes (Fig 2A and B). Next, we determined whether UK5099-mediated increase in mitochondrial respiration was caused by an increase in uncoupled fat oxidation. To this end, we injected ATP-synthase inhibitor oligomycin, to block coupled respiration, and etomoxir, to block fatty acid oxidation, sequentially in NE-stimulated brown adipocytes. MPC inhibition via UK5099 treatment increased both uncoupled and coupled (ATP-synthesizing) respiration, where the effect on coupled respiration was twice as large as the effect on uncoupled respiration (Fig 2A and B). Furthermore, the component of NE-stimulated respiration sensitive to etomoxir was significantly higher in cells treated with UK5099 compared to vehicle-treated cells. Altogether, these data indicate that MPC inhibition promotes an increase in both

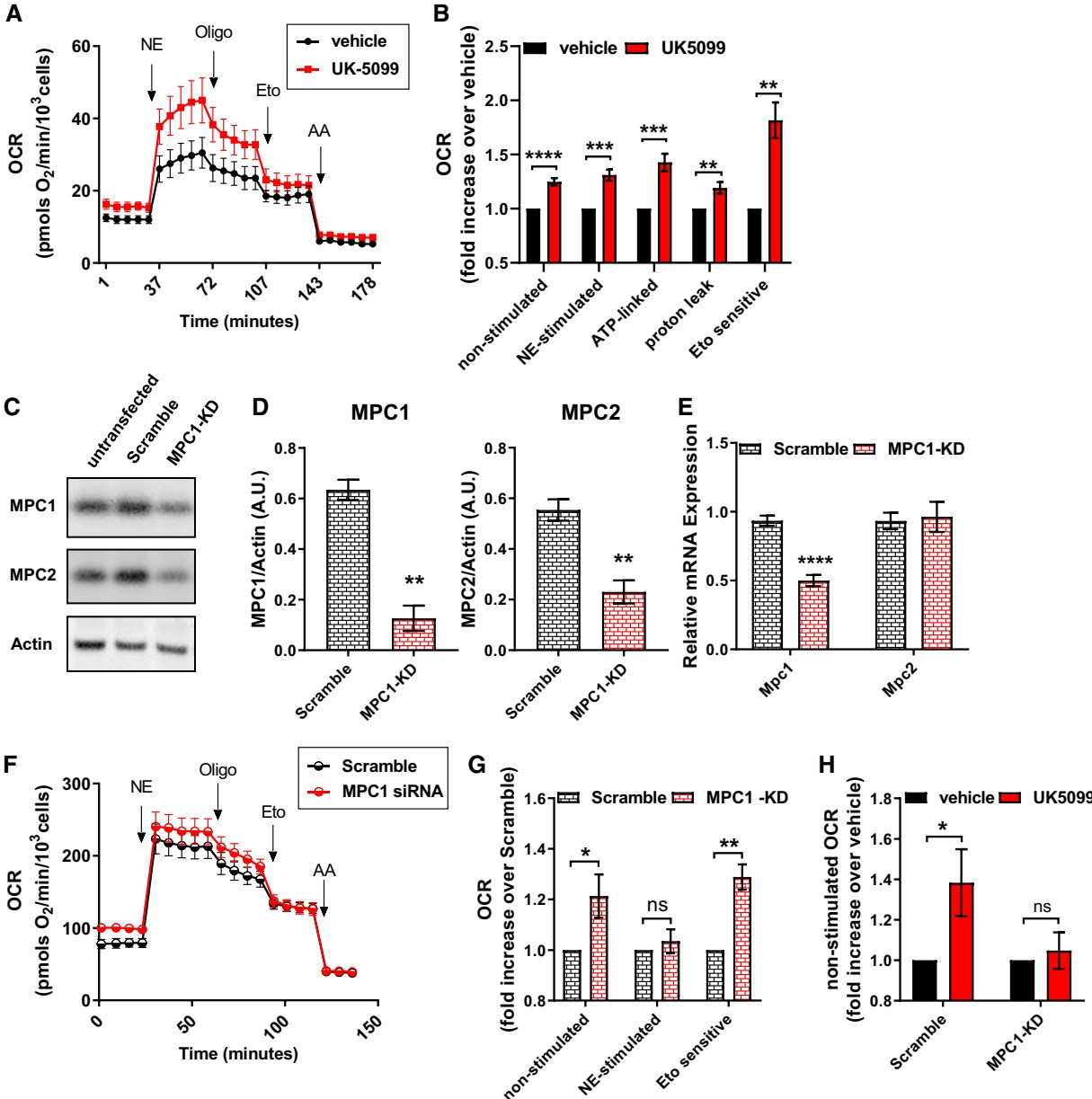

**Figure 2. MPC1 inhibition stimulates mitochondrial respiration coupled to ATP synthesis in brown adipocytes.**

A, B   Effect of UK5099 treatment on norepinephrine-stimulated energy expenditure. Fully differentiated primary brown adipocytes were pre-treated with vehicle (DMSO) or 100 nM UK5099 for 2 h. OCR were measured in respirometry media supplemented with 5 mM glucose and 3 mM glutamine in the presence of vehicle or UK5099. Norepinephrine (NE; 1 μM), oligomycin A (Oligo; 4 μM), etomoxir (Eto; 40 μM), and antimycin A (AA; 4 μM) were injected where indicated. (A) Representative OCR traces averaging 4 technical replicates. (B) Quantification of non-stimulated OCR, NE-stimulated OCR, ATP-linked OCR, mitochondrial proton leak and Eto-sensitive OCR as shown in (A) from $n = 9$ individual experiments. The effects of UK5099 treatment were normalized to vehicle for each experiment. **$P < 0.01$, ***$P < 0.001$, ****$P < 0.0001$ compared to vehicle by Student's *t*-test.

C–H   Primary brown adipocytes were transfected with MPC1 siRNA (MPC1 KD) or Scramble RNA (Scramble). (C) Representative Western blot analysis of MPC1, MPC2, and actin. (D) Quantification of MPC1 and MPC2 expression normalized to actin from Western blots in (C) from $n = 3$ individual experiments. **$P < 0.01$ compared to Scramble by Student's *t*-test. (E) mRNA levels of MPC1 and MPC2 from $n = 3$ individual experiments. ****$P < 0.0001$ compared to Scramble by Student's *t*-test. (F) Representative OCR traces of differentiated primary brown adipocytes averaging 6 technical replicates. OCR were measured in respirometry media supplemented with 5 mM glucose and 3 mM glutamine. Norepinephrine (NE; 1 μM), oligomycin A (Oligo; 4 μM), etomoxir (Eto; 40 μM), and antimycin A (AA; 4 μM) were injected where indicated. (G) Quantification of basal, NE-stimulated and Eto-sensitive OCR as shown in (F) from $n = 7$ individual experiments. Data were normalized to Scramble RNA for each individual experiment. ns $P > 0.05$, *$P < 0.05$, **$P < 0.01$ compared to Scramble by Student's *t*-test. (H) Quantification of basal OCR in response to 100 nM UK5099 treatment in Scramble RNA of MPC1 siRNA-transfected cells from $n = 5$ individual experiments. Data were normalized to vehicle for each experiment. Note that UK5099 capacity to increase OCR in non-stimulated brown adipocytes is absent in cells where MPC1 is downregulated. ns $P > 0.05$, *$P < 0.05$ compared to vehicle by Student's *t*-test.

Data information: All data are presented as mean ± SEM.

coupled and uncoupled fatty acid oxidation during adrenergic stimulation (Fig 2A and B). To address the possibility that UK5099 increased coupled respiration by decreasing UCP1 content, we measured UCP1 expression in cells treated with UK5099. Western blot and qPCR analyses showed that UK5099 treatment did not affect UCP1 expression and protein content (Fig EV2A).

MPC is a protein heterodimer composed of MPC1 and MPC2 subunits, which are required for mitochondrial pyruvate import and consequently pyruvate oxidation (Herzig *et al*, 2012; Bricker *et al*, 2012; Vigueira *et al*, 2014). To validate that the effects on respiration observed were not off-target effects of UK5099, we silenced the expression of *Mpc1* and then assessed energy expenditure and fatty acid oxidation. Brown adipocytes were transfected with an siRNA for *Mpc1* (MPC1-KD), or a control scramble RNA (Scramble). Successful knock-down was confirmed by Western blot (Fig 2C and D) and by qPCR (Fig 2E). Both MPC1 and MPC2 protein levels were reduced when cells were transfected with MPC1 siRNA (Fig 2D). However, only the transcript levels of *Mpc1*, but not *Mpc2*, were reduced in cells transfected with *Mpc1* siRNA (Fig 2E). As MPC1 stabilizes MPC2 by being in the same complex, similar effects of downregulation of MPC2 protein levels by the silencing of MPC1 were reported in previous studies (Bricker *et al*, 2012; Divakaruni *et al*, 2013; Gray *et al*, 2015). Next, we tested the effects of MPC1-KD on energy expenditure and fatty acid oxidation. Assessment of OCR in non-stimulated brown adipocytes showed that MPC1-KD increased energy expenditure compared to control, thereby confirming the observed effect of pharmacological MPC inhibition (Fig 2F and G). Furthermore MPC1-KD increased the etomoxir-sensitive proportion of NE-stimulated respiration in brown adipocytes, indicating that genetic downregulation of *Mpc1* recapitulated the effects of UK5099 (Fig 2G). Interestingly, MPC1-KD did not further increase NE-stimulated OCR when compared to scramble (Fig 2G). The apparent difference between the effects of pharmacological and genetic interference on NE-induced energy expenditure suggests that a long-term reduction of MPC activity recruits compensatory mechanisms to control the thermogenic response to NE. To further confirm that the effects of UK5099 on increased fat oxidation are a consequence of MPC1 blockage, we measured the effects of 100 nM UK5099 on mitochondrial respiration in scramble RNA and MPC1-KD brown adipocytes. UK5099 treatment increased mitochondrial respiration in unstimulated brown adipocytes expressing scramble RNA, similar to the effect observed in non-transfected cells (Fig 2A, B and H). However, this stimulatory effect of UK5099 on OCR was lost in MPC1-KD cells (Fig 2H), indicating that increased energy expenditure caused by UK5099 treatment is dependent on MPC. As in UK5099 treated brown adipocytes, qPCR analysis confirmed that MPC1 KD did not reduce UCP1 expression or affect brown adipocyte differentiation markers compared to control cells (Fig EV2B). Overall, our data suggest that MPC inhibition further increases NE-stimulated energy expenditure and mitochondrial respiration coupled to ATP synthesis.

## Glutamine metabolism is required for increased energy expenditure induced by MPC inhibition

Fatty acid oxidation generates acetyl-CoA (2 carbons) which then condenses with oxaloacetate (4 carbons) to generate citrate (6

carbons) through the citrate synthase reaction. Since this metabolic reaction requires both substrates, the two carbons provided by acetyl-CoA must be matched with oxaloacetate, which is provided by anaplerotic reactions from pyruvate or amino acid metabolism (Cannon & Nedergaard, 1979). As we observed that MPC inhibition increased fatty acid oxidation (Figs 1F and G, and 2A, B, F and G), we hypothesized that brown adipocytes switch to an alternative source for oxaloacetate when mitochondrial pyruvate transport is limited. Previous work in hepatocytes, neurons, and myoblasts showed that cells shift toward glutamate metabolism when the MPC is inhibited, which can provide carbons for oxaloacetate production (Vacanti *et al*, 2014; Yang *et al*, 2014; Divakaruni *et al*, 2017). We therefore assessed the dependency of UK5099-stimulated increase in mitochondrial respiration on glutamine in brown adipocytes. Glutamine was required for UK5099-mediated increase in OCR in both non-stimulated and NE-stimulated brown adipocytes (Fig 3A and B). In media lacking glutamine, pharmacological inhibition of the MPC had no effect on non-stimulated OCR and reduced NE-stimulated respiration (Fig 3A and B). Furthermore, use of glutamine-depleted media prevented the stimulatory effects of UK5099 on both coupled and uncoupled respiration, as well as etomoxir-sensitive respiration (Fig 3A and B). These data indicate that glutamine is a necessary nutrient to sustain the increase in mitochondrial fat oxidation induced by MPC inhibition (Fig 3A and B). Glutamine catabolism is a source for oxaloacetate, which is necessary for acetyl-CoA generated by fatty acid oxidation to enter the TCA cycle. Thus, to test whether glutamine catabolism was necessary for UK5099-mediated increase in mitochondrial fat oxidation, we tested the effects of glutaminase inhibitor CB839 on UK5099-induced mitochondrial fat oxidation. Figure 3C shows that CB839 prevented the increases in mitochondrial coupled and uncoupled respiration and etomoxir-sensitive OCR caused by MPC inhibition, supporting our hypothesis that glutaminolysis is required to support enhanced mitochondrial fatty acid oxidation induced by MPC inhibition (Fig 3C). One pathway to fuel the TCA cycle with glutamine is through the glutamic oxaloacetic transaminase (GOT) reaction, a reaction with converts glutamate (from glutamine) and oxaloacetate into alpha-ketoglutarate (used by the TCA cycle) and aspartate. Therefore, we reasoned that the aspartate:glutamate ratio would serve as an indicator of glutamine oxidation by the TCA cycle (Vacanti *et al*, 2014; Divakaruni *et al*, 2017). Indeed, we find that UK5099 treatment increased aspartate:glutamate ratio indicating increased GOT activity (Fig 3D). To specifically determine the metabolic fate of glutamine into the TCA cycle intermediates, we traced the incorporation of glutamine carbons to polar metabolites, using uniformly labeled glutamine [U-$^{13}$C$_5$] and quantifying labeled metabolite enrichment by GC-MS (Fig EV3B). UK5099 significantly increased the incorporation of glutamine-derived carbons into aspartate, glutamate, α-ketoglutarate and malate, thereby supporting the hypothesis that glutamine is required for TCA anaplerosis, when mitochondrial pyruvate import is limited (Fig EV3C). Intriguingly, the isotopomer distribution shows that most of the increase in glutamine-derived carbons were in the M + 3 mass isotopomer (Fig EV3D), which may be indicative of reductive carboxylation of glutamine linked to ATP citrate lyase activity (Zhang *et al*, 2014).

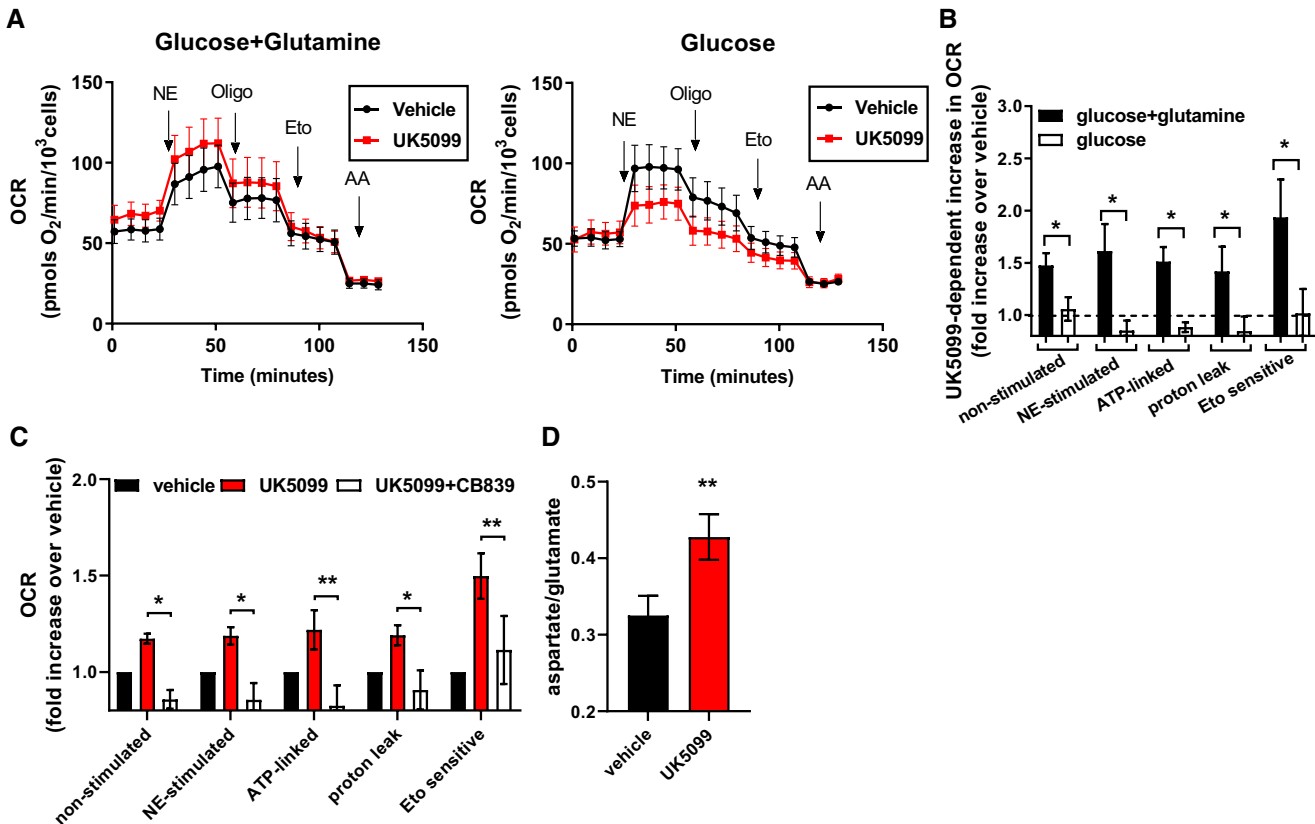

**Figure 3. Glutamine metabolism is required for increased energy expenditure induced by MPC inhibition.**

A, B    Requirement for glutamine in UK5099-induced energy expenditure. Primary brown adipocytes were pre-treated with vehicle (DMSO) or 100 nM UK5099 for 2 h.
          OCR were measured in presence of either 5 mM glucose and 3 mM glutamine or 5 mM glucose alone. Norepinephrine (NE; 1 μM), oligomycin A (Oligo; 4 μM),
          etomoxir (Eto; 40 μM), and antimycin A (AA; 4 μM) were injected where indicated. (A) Representative OCR traces averaging 6 technical replicates. (B) Quantification
          of non-stimulated OCR, NE-stimulated OCR, ATP-linked OCR, mitochondrial proton leak and etomoxir-sensitive OCR after vehicle or UK5099 treatment as shown in
          (A) from $n = 4$ individual experiments. Data were normalized to vehicle for each experiment. Note that UK5099 treatment increased OCR only when cells were
          assayed in the presence of both glucose and glutamine. *$P < 0.05$, compared to vehicle by Student's $t$-test.
C       Effect of glutaminase inhibitor on UK5099-induced energy expenditure. Primary brown adipocytes were pre-treated with either vehicle (DMSO), 100 nM UK5099, or
          100 nM UK5099 + 2.5 μM CB839 for 2 h. OCR were measured in respirometry media supplemented with 5 mM glucose and 3 mM glutamine in the presence of
          vehicle, UK5099 or UK5099 + CB839 from $n = 5$ individual experiments. Data were normalized to vehicle for each experiment. *$P < 0.05$, **$P < 0.01$ by ANOVA.
D       Effect of MPC inhibition on glutamate/aspartate ratio measurement. Glutamate/aspartate ratio was quantified using GC-MS as a measure of glutamine catabolism.
          Primary brown adipocytes were treated for 24 h with vehicle (DMSO) or 10 μM UK5099 from $n = 6$ individual experiments. **$P < 0.01$, compared to vehicle by
          Student's $t$-test.

Data information: All data are presented as mean ± SEM.

## The malate-aspartate shuttle supports the increase of both glutamine and fatty acid metabolism induced by MPC inhibition

Glutamic oxaloacetic transaminase is a critical component of the malate-aspartate shuttle (MASh), a cyclic pathway that allows the transfer of reduced equivalents from the cytosol to the mitochondrial matrix (Fig 4A). Since our metabolomics data indicated increased GOT activity upon UK5099 treatment (Fig 3D), we sought to test the role of the MASh in UK5099-induced energy expenditure in brown adipocytes. To determine whether MASh could be contributing to the increase in respiration induced by MPC inhibition, we inhibited GOT using aminooxyacetic acid (AOA) (Kauppinen *et al*, 1987). The increase in mitochondrial respiration induced by UK5099 was dose-dependently reduced by AOA, reaching complete inhibition at 1 mM AOA (Figs 4B and EV4A). In the

absence of UK5099, AOA caused no apparent effects on brown adipocyte respiration (Figs 4B and EV4A). To further confirm the involvement of the MASh in the metabolic effects caused by UK5099, and address potential off-target effects of AOA (Cornell *et al*, 1984; Yang *et al*, 2008), we then silenced the expression of oxoglutarate carrier 1 (*SLC25A11* or *OGC1*), a key component of MASh. OGC1 mediates the electroneutral exchange of malate and α-ketoglutarate between mitochondrial intermembrane space and the matrix, transporting malate to mitochondrial matrix to generate oxaloacetate (Fig 4A) (Indiveri *et al*, 1987). Adenovirus-mediated knock-down of *OGC1* was confirmed by qPCR in primary brown adipocytes (Fig EV4B). OGC1 KD prevented the UK5099-induced increase in OCR in non-stimulated and NE-stimulated brown adipocytes (Figs 4C and D, and EV4C). Furthermore, OGC1 KD abrogated the increase in etomoxir-sensitive OCR caused by UK5099 (Fig 4E).

The involvement of the MASh in UK5099-induced energy expenditure was further assessed by silencing another MASh component, the mitochondrial aspartate/
glutamate carrier (*SLC25A12* or *Aralar1*). Aralar1 catalyzes the calcium-dependent exchange of aspartate efflux and glutamate

uptake across the mitochondrial inner membrane (Palmieri *et al*, 2001). Aralar is found as two isoforms 1 and 2 (*Aralar1* or *SLC25A12,* and *Aralar2* or citrin or *SLC25A13*), with Aralar1 being the most abundant isoform in BAT (Forner *et al*, 2009; Geiger *et al*, 2013). siRNA-mediated knock-down of *Aralar1* was confirmed by

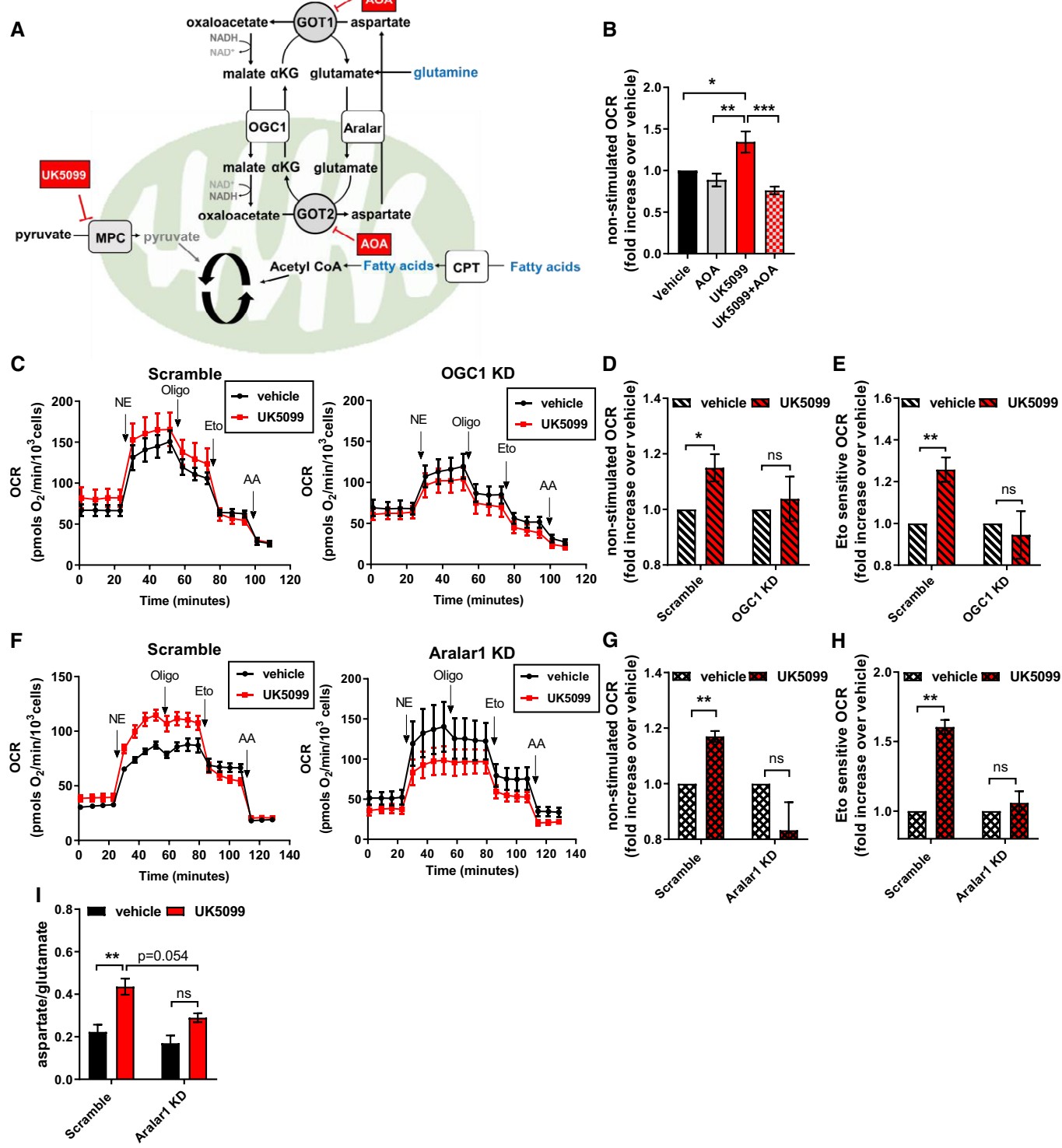

**Figure 4.**

◀

**Figure 4. The malate-aspartate shuttle supports the increase of both glutamine and fatty acid metabolism induced by MPC inhibition.**

A Schematic representation of proposed mechanism by which MPC inhibition engages the malate-aspartate shuttle (MASh). GOT1/2, glutamic oxaloacetic transaminase; OGC1, oxoglutarate carrier 1; Aralar, mitochondrial aspartate-glutamate carrier; CPT, Carnitine Palmitoyltransferase. The coordinated work of OGC1 and Aralar provides a mechanism to allow fatty acid oxidation in the absence of pyruvate import into mitochondria. Fatty acid oxidation generates acetyl-CoA which condenses with oxaloacetate to generate citrate. In the absence of pyruvate import into mitochondria, oxaloacetate can be generated by glutamine metabolism.

B Effect of transaminase inhibition on UK5099-induced energy expenditure. Brown adipocytes were pre-treated with vehicle (DMSO), 50–100 nM UK5099, 1 mM aminooxyacetic acid (AOA), or a combination of UK5099 and AOA for 2 h. OCR were measured in respirometry media supplemented with 5 mM glucose and 3 mM glutamine in the presence of the tested compounds. Data show quantification of non-stimulated OCR from $n = 5$ individual experiments. Data were normalized to vehicle for each experiment. *$P < 0.05$, **$P < 0.01$, ***$P < 0.001$ by ANOVA.

C–E Effect OGC1 downregulation on UK5099-induced energy expenditure. Primary brown adipocytes were transduced with Scramble RNA (Scramble) or shOGC1 (OGC1 KD) adenovirus. Cells were pre-treated for 2 h with vehicle (DMSO) or 100 nM UK5099 before OCR measurements. OCR were measured in respirometry media supplemented with 5 mM glucose and 3 mM glutamine in the presence of vehicle or UK5099. Norepinephrine (NE; 1 µM,) oligomycin A (Oligo; 4 µM), etomoxir (Eto; 40 µM), and antimycin A (AA; 4 µM) were injected where indicated. (C) Representative OCR traces averaging 4 technical replicates. (D) Quantification of non-stimulated OCR as measured in (C) from $n = 6$ individual experiments. Data were normalized to vehicle for each experiment. ns $P > 0.05$, *$P < 0.05$ compared to vehicle by Student's *t*-test. (E) Quantification of Eto-sensitive OCR as measured in (C) from $n = 6$ individual experiments. Data were normalized to vehicle for each experiment. ns $P > 0.05$, **$P < 0.01$ compared to vehicle by Student's *t*-test.

F–H Effect of Aralar1 downregulation on UK5099-induced energy expenditure. Primary brown adipocytes were transfected with Scramble RNA (Scramble) or siRNA for Aralar1 (Aralar1 KD). Cells were pre-treated for 2 h with vehicle (DMSO) or 100 nM UK5099 before OCR measurements. OCR were measured in respirometry media supplemented with 5 mM glucose and 3 mM glutamine in the presence of vehicle or UK5099. Norepinephrine (NE; 1 µM), oligomycin a (Oligo; 4 µM), etomoxir (Eto; 40 µM), and antimycin a (AA; 4 µM) were injected where indicated. (F) Representative OCR traces averaging 4 technical replicates. (G) Quantification of non-stimulated OCR as measured in (G) from $n = 4$ individual experiments. Data were normalized to vehicle for each experiment. ns $P > 0.05$, **$P < 0.01$ compared to vehicle by Student's *t*-test. (H) Quantification of Eto-sensitive OCR as measured in (F) from $n = 4$ individual experiments. Data were normalized to vehicle for each experiment. ns $P > 0.05$, **$P < 0.01$ compared to vehicle by Student's *t*-test.

I Brown adipocytes transfected with Scramble RNA or Aralar1 siRNA were treated for 24 h with vehicle (DMSO) or 10 µM UK5099. Data show quantification of the ratio of aspartate to glutamate abundance as measured by GC-MS from $n = 3$ individual experiments. ns $P > 0.05$, **$P < 0.01$ by two-way ANOVA. Note that both carriers of malate-aspartate shuttle, OGC1 and Aralar1, are required for UK5099-induced energy expenditure.

Data information: All data are presented as mean ± SEM.

qPCR in primary brown adipocytes (Fig EV4D). Similar to OGC1 KD cells, Aralar1 KD reversed the stimulatory effects of MPC inhibition on respiratory rates in non-stimulated and NE-stimulated brown adipocytes (Figs 4F and G, and EV4E). Furthermore, knock-down of *Aralar1* reversed the effects of MPC inhibition on NE-stimulated mitochondrial fat oxidation, as revealed by the lack of an increase in etomoxir-sensitive respiration following UK5099 treatment (Fig 4H).

Next, we sought to determine whether the MASh is required for the increase in glutamine metabolism. To test this hypothesis, we analyzed the effect of UK5099 treatment on glutamate and aspartate abundance in Aralar1 KD as compared to scrambled siRNA treated brown adipocytes. Here we applied the same principle as in Fig 3D, where the ratio of aspartate to glutamate is used as a measure of aspartate aminotransferase activity and associated glutamine catabolism. Knock-down of *Aralar1* resulted in partial reversal of the increase in aspartate:glutamate ratio induced by MPC inhibition (Fig 4I). Partial reversal was expected, given that glutamine can provide glutamate and alpha-ketoglutarate independently of the MASh, through glutaminase and glutamate dehydrogenase. Thus, our data indicate that the Malate-Aspartate Shuttle is required to allow an increase in Energy Expenditure in Brown Adipocytes (MAShEEBA) via glutamate and fatty acid oxidation under MPC inhibition. Having mapped the metabolic reprogramming associated with the increased energy expenditure, we then asked which ATP-requiring processes were increased upon MPC inhibition.

## MPC inhibition induces ATP utilization by lipid cycling

Intracellular handling of nutrients, particularly of fatty acids, is an ATP-demanding process. Our data and other's demonstrated that when mitochondria do not have access to pyruvate, a cellular response elicits a switch to use fatty acids as an oxidative fuel in the mitochondria to synthesize ATP (Vacanti *et al*, 2014; Gray *et al*, 2015). Accordingly, in addition to the increase in mitochondrial fat oxidation induced by MPC inhibition, we find that UK5099 treatment increased ATP-synthesizing respiration both in NE-stimulated (Figs 2A and 3B) and non-stimulated brown adipocytes in a dose-dependent manner (Figs 5A and B, and EV5D). This was accompanied by an increase in mitochondrial proton leak in response to UK5099 treatment (Fig 5A and B), which may be the result of increased membrane potential associated with a shift to fatty acid oxidation and the engagement of complex II (Nicholls & Ferguson, 2013). Analysis of TMRE fluorescence intensity confirmed an increase in mitochondrial membrane potential in response to treatment with UK5099 (Fig EV5A and B). To confirm that increase TMRE fluorescence is not due to increased cytosolic concentration of TMRE, we show that TMRE fluorescence intensity in the nuclear area, which represents a non-mitochondrial TMRE level in the cell, is unchanged (Fig EV5C).

We reasoned that if ATP demand is elevated by MPC inhibition, then UK5099 treatment should have no effect on mitochondrial respiration when cellular ATP demand is clamped, by controlling cytoplasmic ADP concentrations. To test this hypothesis, we clamped the ADP concentration by permeabilizing brown adipocytes and supplementing the assay buffer with 5 mM ADP. OCR was measured after concurrently providing all fuels (pyruvate, malate, palmitoyl-CoA and carnitine). UK5099 treatment did not change mitochondrial respiration in permeabilized cells when given all substrates and 5 mM ADP (Fig 5C). This observation strengthens the finding that increased mitochondrial respiration in intact brown adipocytes induced by MPC inhibition is mainly driven by increased ATP demand.

Brown and white adipocytes re-esterify free fatty acids that are produced by lipolysis, in a process termed glycerolipid and free fatty

acid (GL/FFA) cycling or lipid cycling (Fig 5D) (Gorin *et al*, 1969; Hammond & Johnston, 1987; Prentki & Madiraju, 2008; Mottillo *et al*, 2014). This ATP-consuming process is inevitable in cells storing and mobilizing fatty acids. Indeed, lipid cycling contributes to the basal ATP demand and is a key mechanism for preventing toxicity caused by excessive levels of free fatty acids and glucose (Prentki & Madiraju, 2008; Mottillo *et al*, 2014). A metabolic signature of lipid cycling is the diversion of glycolytic intermediates toward glycerol-3-phosphate (G3P) production to support FFA re-esterification into triglycerides (Reshef *et al*, 2003). Since MPC inhibition activated lipolysis and fatty acid oxidation (Fig 1), we hypothesized that increased lipid cycling could explain the increased ATP demand observed upon UK5099 treatment in brown adipocytes. If this were the case, G3P/DHAP would be increased by MPC inhibition. Indeed,

the G3P/DHAP ratio was increased in brown adipocytes treated with UK5099 (Fig 5D and E), via an increase in G3P, rather than a decrease in DHAP content (Fig EV5E). We then reasoned that inhibition of lipolysis or lipid re-esterification should prevent UK5099-induced increase in ATP demand. Figure 5F shows that specific inhibition of adipose triglyceride lipase (ATGL) by Atglistatin, which impairs the first step of lipolysis, completely prevented the increase in basal and ATP-synthesizing respiration induced by MPC inhibition in non-stimulated brown adipocytes (Fig 5F and G). Remarkably, Atglistatin treatment had no significant effects on maximal (uncoupled) respiratory rates in intact brown adipocytes (Fig EV5F), underscoring the concept that increased ATP demand associated with lipolysis drives the increase in respiration induced by MPC inhibition.

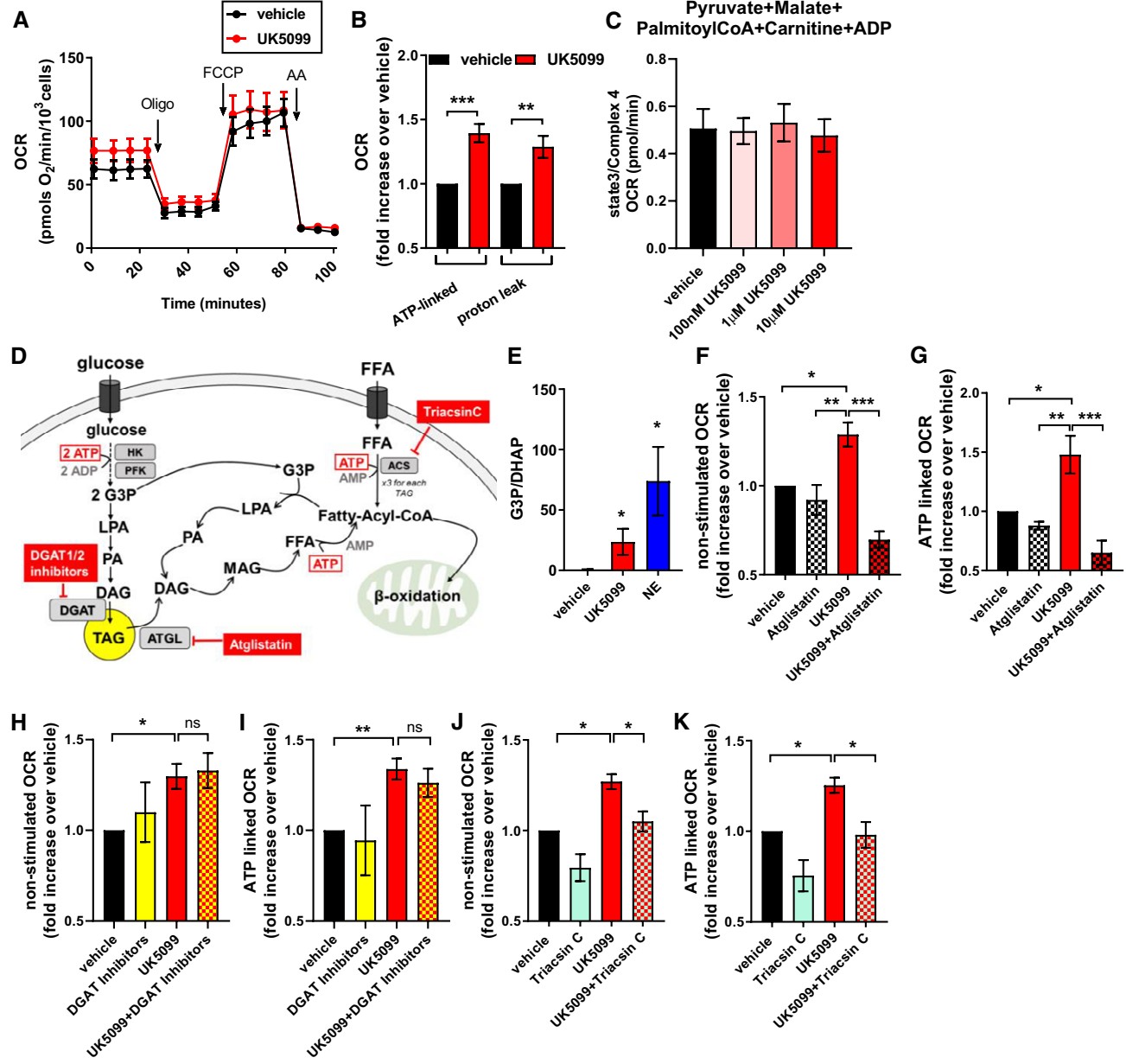

**Figure 5.**

**Figure 5. MPC inhibition induces ATP utilization by lipid cycling.**

A, B Effect of UK5099 on energy expenditure contributed by ATP demand and by mitochondrial proton leak in non-stimulated brown adipocytes. Primary brown adipocytes were pre-treated with vehicle (DMSO) or 100 nM UK5099 for 2 h. OCR were measured in respirometry media supplemented with 5 mM glucose and 3 mM glutamine in the presence of vehicle or UK5099. Oligomycin A (Oligo; 4 μM), mitochondrial uncoupler FCCP (2 μM) and antimycin A (AA; 4 μM) were injected where indicated. (A) Representative OCR traces averaging 6 technical replicates. (B) Quantification of mitochondrial proton leak and ATP-linked OCR (oligomycin sensitive) as measured in (A) from $n = 9$ individual experiments. Data were normalized to vehicle for each individual experiment. **$P < 0.01$ ***$P < 0.001$ compared to vehicle by Student's $t$-test.

C To determine the contribution of ATP demand to increased energy expenditure under UK5099, cells were permeabilized and cytosolic ADP concentrations were clamped. OCR were measured in the presence of 5 mM pyruvate, 3 mM malate, 0.1 mM palmitoyl-CoA, 0.5 mM carnitine, and 5 mM ADP. Cells were treated with either 100 nM, 1 μM or 10 μM UK5099. Data show maximal state 3 OCR normalized to maximal complex 4 activity from $n = 3$ individual experiments. Note that when ATP demand is clamped, UK5099 had no effect on energy expenditure.

D Schematic representation of lipid cycling. HK, hexokinase; PFK, phosphofructokinase; ATGL, adipose triglyceride lipase; ACS, Acyl-CoA synthetase; TAG, triacylglyceride; DAG, diacylglyceride; MAG, monoacylglyceride; LPA, lysophosphatidic acid; FFA, free fatty acid; G3P, glycerol-3-phosphate

E Effect of MPC inhibition on glycerol 3-phosphate (G3P) and dihydroxyacetone phosphate (DHAP) generation. G3P/DHAP is used as a measure of acylglycerol synthesis. Brown adipocytes were treated with either vehicle, 100 nM UK5099 or 1 μM norepinephrine (NE) for 24 h from $n = 4$ individual experiments. *$P < 0.05$ by ANOVA, relative to vehicle.

F, G Contribution of lipolysis to UK5099 induced energy demand. Primary brown adipocytes were pre-treated with either vehicle (DMSO), 100 nM UK5099, 40 μM Atglistatin or a combination of Atglistatin with UK5099 for 2 h. OCR were measured in respirometry media supplemented with 5 mM glucose and 3 mM glutamine in the presence of vehicle, UK5099, or Atglistatin. (F) Quantification of non-stimulated OCR from $n = 4$ individual experiments. *$P < 0.05$, **$P < 0.01$, ***$P < 0.001$ by ANOVA. (G) Quantification of ATP-linked respiration from $n = 4$ individual experiments. Data were normalized to vehicle for each individual experiment. *$P < 0.05$, **$P < 0.01$, ***$P < 0.001$ by ANOVA.

H, I Contribution of TAG synthesis to UK5099 induced energy demand. Primary brown adipocytes were pre-treated with either vehicle (DMSO), 100 nM UK5099, 1 μM DGAT1, and 1 μM DGAT2 or a combination of DGAT1/2 inhibitors with UK5099 for 2 h. OCR were measured in respirometry media supplemented with 5 mM glucose and 3 mM glutamine in the presence of vehicle, UK5099, DGAT1/2 inhibitors, or UK5099 + DGAT1/2 inhibitors. (H) Quantification of non-stimulated OCR from $n = 6$ individual experiments. Data were normalized to vehicle for each individual experiment. ns $P > 0.05$, *$P < 0.05$ by ANOVA. (I) Quantification of ATP-linked OCR from $n = 6$ individual experiments. Data were normalized to vehicle for each individual experiment. ns $P > 0.05$, **$P < 0.01$ by ANOVA.

J, K Contribution of ACS to UK5099 induced energy demand. Primary brown adipocytes were pre-treated with either vehicle (DMSO), 100 nM UK5099, 5 μM Triacsin C or a combination of 5 μM Triacsin C with UK5099 for 2 h. OCR were measured in respirometry media supplemented with 5 mM glucose and 3 mM glutamine in the presence of vehicle, UK5099, Triacsin C, or UK5099 + Triacsin C. Quantification of non-stimulated OCR from $n = 4$ individual experiments. Data were normalized to vehicle for each individual experiment. *$P < 0.05$ by ANOVA. Quantification of ATP-linked OCR from $n = 4$ individual experiments. Data were normalized to vehicle for each individual experiment. *$P < 0.05$ by ANOVA. Note that MPC inhibition-induced energy demand requires lipolysis and ACS-dependent fatty acid activation, but not DGAT-dependent TAG synthesis, suggesting the activation of a sub-cycle in lipid cycling pathway.

Data information: All data are presented as mean ± SEM.

Lipid cycling can involve the entire cycle from TAG synthesis to free fatty acid and glycerol or be limited to shorter sub-cycles that involve break-down and regeneration of mono- and di-glycerides. Importantly, full or partial cycling are both ATP-demanding processes (Prentki & Madiraju, 2008). To assess the potential role of TAG synthesis in UK5099-mediated activation of lipid cycling, we blocked the last step of TAG synthesis catalyzed by diglyceride acyltransferase 1 and 2 (DGAT1/2) using the pharmacological inhibitors JNJ-DGAT1-A (Qi et al, 2010) and PF-06424439. The UK5099-induced increases in respiration were insensitive to both DGAT inhibitors (Figs 5H and I). These results suggest that ATP demand induced by MPC inhibition does not involve the last step of TAG synthesis. We then blocked acyl-CoA synthetase (ACS) by using Triacsin C, thereby blocking a sub-cycle of lipid cycling (Fig 5D). The UK5099-induced increase in ATP-synthesizing respiration was highly sensitive to Triacsin C, reaching values close to vehicle-treated cells (Fig 5J and K). In the absence of UK5099, Triacsin C reduced basal and ATP-synthesizing respiration in non-stimulated brown adipocytes, in agreement with previous studies showing that lipid esterification contributes to basal ATP demand in adipose tissue (Mottillo et al, 2014). We confirmed that Triacsin C and UK5099 treatment did not affect maximal (uncoupled) respiratory rates, strengthening that inhibition of ACS only decreases mitochondrial ATP demand and does not affect mitochondrial fuel availability and respiratory function (Fig EV5G). Collectively, our data suggest that MPC inhibition increases ATP demand and OCR in non-stimulated brown adipocytes by activating a sub-cycle of lipid cycling pathway.

# Discussion

Our study identified that inhibition of mitochondrial pyruvate transport is a novel mechanism to increase the energy demand and expenditure in non-stimulated brown adipocytes through the activation of lipid cycling (Fig 6). Non-stimulated brown adipocytes store most of their intracellular lipids as TAGs in lipid droplets, which are consumed upon adrenergic stimulation (Cannon & Nedergaard, 2004). Therefore, increasing fat oxidation in non-stimulated BAT, and potentially other types of adipose tissues, could be a promising way to reduce circulating fatty acids and shrink fat depots. Our data suggest that MPC inhibition increases lipolysis and fatty acid oxidation in both non-stimulated and NE-stimulated brown adipocytes. Fatty acid oxidation was assessed by measuring [U-$^{13}$C$_{16}$] palmitate incorporation into TCA cycle metabolites and using the CPT1 inhibitor etomoxir in respirometry assays. As etomoxir has been reported to have off-target effects particularly at higher doses (Divakaruni et al, 2018), we confirmed that the dose of etomoxir used in this study does not affect the oxidation of substrates that are independent of CPT1 activity (Fig EV1C). Interestingly, a recent study showed that whole body MPC1 KO mice was embryonically lethal, unless the pregnant mice were fed a ketogenic diet (Vanderperre et al, 2016). Further evidence reinforced the shift toward fat oxidation in heterozygous MPC1 KO mice, which exhibit reduced body weight and increased expression of genes associated with lipolysis and fatty acid oxidation (Zou et al, 2018a). Additionally data in C2C12 cells and muscle specific MPC1 KO mice show increased fatty acid oxidation (Vacanti et al, 2014; Sharma et al,

## Brown Adipocytes

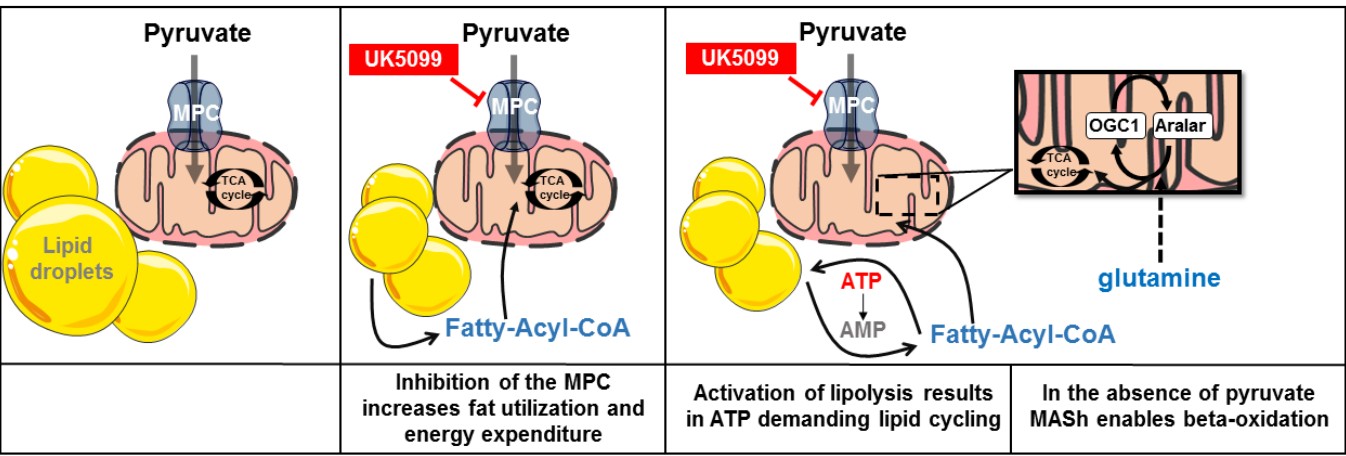

**Figure 6. Proposed mechanism by which MPC inhibition induces lipid cycling and the activation of the malate-aspartate shuttle.**
Inhibition of mitochondrial pyruvate import in brown adipocytes results in the induction of lipolysis, a shift toward fatty acid utilization and an increase in energy expenditure. The increase in ATP demand by is a consequence of the activation of lipid cycling, involving a recurrent cycle of fatty acid release and re-esterification. To support fatty acid oxidation in the absence of pyruvate import, the TCA cycle relies on carbons from glutamine and the activation of the malate-aspartate shuttle. Thus blocking mitochondrial pyruvate import can promote an increase in energy wasting even in the absence of adrenergic stimulation. LD, lipid droplet; MPC, mitochondrial pyruvate carrier; CPT1, Carnitine Palmitoyltransferase1; ATP, adenosine triphosphate; AMP, adenosine monophosphate; OGC1, oxoglutarate carrier 1; Aralar, mitochondrial aspartate-glutamate carrier.

2019). These observations agree with our results showing increased lipid droplet consumption and fatty acid oxidation in brown adipocytes upon MPC inhibition. Our data also indicate that MPC inhibition acutely increases fatty acid oxidation and therefore could be envisioned as a potential target for obesity and associated metabolic disorders by promoting fat oxidation in vivo. Recent studies demonstrated that genetic deletion of mice MPC in UCP1 + adipocytes improved insulin sensitivity and increased circulating ketones, consistent with our data showing increased fat oxidation in isolated brown adipocytes (Panic *et al*, 2020). It remains to be determined whether MPC inhibition can promote fat oxidation and lipid cycling in white adipocytes, which would be especially interesting considering that most depots of human adipose tissue are white or beige.

Multiple groups have demonstrated that modulation of MPC activity acts as a mitochondrial fuel preference switch in various tissues and conditions (Vacanti *et al*, 2014; Yang *et al*, 2014; Gray *et al*, 2015; Divakaruni *et al*, 2017; Schell *et al*, 2017; Zou *et al*, 2018a, 2018b). However, in these studies, pharmacological inhibition, or genetic deletion of the MPC led to reduced or unchanged mitochondrial respiration. By contrast, in brown adipocytes MPC inhibition increased non-stimulated and adrenergically stimulated mitochondrial oxygen consumption. One possible reason that could explain this discrepancy is that the brown adipocyte is a cell type specialized to store and oxidize fat. Therefore, it is conceivable that brown adipocytes are more efficient at switching their respiratory fuel to fatty acids when mitochondrial pyruvate import is limited. Enhanced fatty acid oxidation and fatty acid availability to mitochondria is often associated with an increase in mitochondrial proton leak through UCP1 activation (Fedorenko *et al*, 2012). Surprisingly, we show that MPC inhibition indeed activated mitochondrial fatty acid oxidation coupled to mitochondrial ATP synthesizing respiration. We find that UK5099 increases ATP-linked OCR

even in brown adipocytes stimulated with NE, suggesting that MPC inhibition can increase fat oxidation to produce ATP in mitochondria in which UCP1 was not successfully activated. In this regard, UCP1 expression varies between different brown adipocytes, and even between different mitochondria inside the same cell (Wikstrom *et al*, 2014; Benador *et al*, 2018). Interestingly the effect of knockdown of MPC1, while being similar to effects of UK5099-treatment in the absence of NE, did not reproduce the same effect in NE-stimulated brown adipocytes. The difference between the effects of pharmacological and genetic interference on NE-induced energy expenditure may be attributed to differences in level of MPC inhibition between UK5099 and genetic knock-down of MPC1 or recruitment of compensatory mechanisms in MPC1 knock-down cells that are not induced upon 2 h of UK5099 treatment.

We show that MPC in brown adipocytes increases energy expenditure through the activation of ATP-demanding lipid cycling. Using pharmacological inhibitors of various steps in TAG break-down or esterification, we show that ATGL-dependent lipolysis and ACS-dependent fatty acid esterification are required for the increased energy demand induced by MPC inhibition (Fig 5F–K). Remarkably we find that inhibition of DGAT enzymes had no effect on energy demand under UK5099 treatment (Fig 5H and I). These data suggest that MPC inhibition activates a sub-cycle in the lipid cycling pathway which involves ACS-dependent esterification of lipids. Furthermore, our data suggest that the majority of the lipids used in the sub-cycle are from endogenous TAG stores that are released from LDs in an ATGL-dependent manner. However, it is conceivable that DGAT inhibition using pharmacological inhibitors was incomplete and therefore had no apparent effect on energy demand following MPC inhibition. It remains to be determined by which mechanism inhibition of the MPC stimulates lipolysis and lipid cycling. In this regard, ROS activate lipolysis in white adipocytes (Krawczyk *et al*,

2012) and blocking MPC decreases the synthesis of the ROS-scavenger glutathione, by diverting glutamine away from glutathione biosynthesis (Tompkins *et al*, 2019). Thus, it is a possibility that lipolysis and the concomitant lipid cycling are initiated by redox-signaling induced by MPC blockage.

Our data support that glycolytic intermediates are diverted toward lipid cycling, instead of downstream glycolysis to pyruvate and lactate, as G3P/DHAP ratio increased upon MPC inhibition (Figs 5E and EV5B). As opposed to increasing energy wasting by mitochondrial uncoupling, activation of lipid cycling would be a safer and more controlled way to increase energy wasting. Recent publications proposed UCP1-independent futile cycles as possible mechanisms to increase energy expenditure in adipose tissue (Kazak *et al*, 2015; Ikeda *et al*, 2017; Mahdaviani *et al*, 2017; Deng *et al*, 2018). Future studies will determine whether lipid cycling can be activated by MPC inhibition in white and beige adipose tissue as well.

In addition to increased oxidation of fatty acids, our data indicate that glutamine supports energy expenditure induced by MPC inhibition, as previously shown in other tissues (Vacanti *et al*, 2014; Yang *et al*, 2014; Divakaruni *et al*, 2017). Besides the increase in fatty acid oxidation and glutamine catabolism to provide oxaloacetate and TCA intermediates, we find that MPC inhibition in brown adipocytes requires the MASh carriers to increase mitochondrial respiration by transferring more NADH from the cytosol to the mitochondria. Furthermore, our glutamine-tracing experiments suggest reductive carboxylation of glutamine, which could be a mechanism for brown adipocytes to sustain *de novo* lipogenesis despite reduced mitochondrial pyruvate import (Yoo *et al*, 2008; Zhang *et al*, 2014). Therefore, the identification that **MASh** supports **E**nergy **E**xpenditure in **B**rown **A**dipocytes (or MAShEEBA) represents a novel mechanism for regulation of energy expenditure in BAT. Wang *et al* showed that acetyl-CoA derived from fatty acids can acetylate and thereby activate components of the MASh, which could be the mechanism by which MPC inhibition increases MASh flux (Wang *et al*, 2018). Intriguingly the only described post-translational modification of the MPC is acetylation of MPC2, which was shown to have an inhibitory effect on MPC activity (Vadvalkar *et al*, 2017). The observation that acetylation has activating effects on the MASh, and inhibitory effects on the MPC might suggest a functional link between MASh activity and MPC activity. However, the functional role of MASh under physiological stimulation of non-shivering thermogenesis in BAT remains to be determined.

In conclusion, we identified a novel mechanism to activate futile lipid cycling and increase energy expenditure in brown adipocytes through inhibition of the MPC. Importantly, FDA-approved drugs such as thiazolidinediones (TZDs) were shown to target the MPC at clinically relevant concentrations (Divakaruni *et al*, 2013). Remarkably, TZDs were shown to promote glyceroneogenesis and lipid cycling through increased expression of glycerol kinase (GK) in human adipose tissue (Tan *et al*, 2003). However, this effect was a result of PPARγ mediated activation of GK transcription, and therefore can likely not be attributed to TZDs effect on the MPC. Further work is required to assess the contribution of lipid cycling and the MASh in the beneficial effects of TZDs. Nevertheless, our data suggests that the MPC could be a safe target to increase energy expenditure in brown adipocytes and potentially improve whole body metabolic health in patients with obesity and cardiometabolic disorders linked to nutrient excess.

# Materials and Methods

## Animals

Primary brown adipocytes were isolated from 4 to 5 weeks old wild-type male C57BL/6J mice (Jackson Laboratory, Bar Harbor, ME). Animals were fed standard chow (mouse diet 9F, PMI Nutrition International, Brentwood, MO) and maintained under controlled conditions (19–22°C and a 14:10 h light-dark cycle) until euthanasia by isofluorane, followed by cervical dislocation. All animal procedures were performed in accordance with the Guide for Care and Use of Laboratory Animals of the NIH and were approved by the ARC/IACUC of the University of California, Los Angeles.

## Primary brown adipocyte culture

Primary brown adipocytes were isolated and cultured as described in detail (Cannon & Nedergaard, 2001). In brief, BAT was dissected from interscapular, subscapular, and cervical regions of four male mice. Tissue was digested using Collagenase Type II (Worthington, Lakewood, NJ). Digested tissue was filtered through a 100 and 40 μm mesh and centrifuged. Cell pellet was re-suspended in brown adipocyte culture media (DMEM supplemented with 10 % newborn calf serum (Sigma-Aldrich, St. Louis, MO), 4 mM glutamine, 10 mM HEPES, 0.1 mg/ml sodium ascorbate, 100 U/ml penicillin, 100 μ/ml streptomycin and plated in a 6-well plate. Cells were incubated in a 37°C, 8% $CO_2$ incubator. 72 hours after isolation the cells were lifted using STEMPro Accutase (Thermo Fisher Scientific, Roskilde, Denmark), counted, and re-plated in final experimental vessel. 24 hours later media was changed to differentiation media (growth media supplemented with 1 μM rosiglitazone maleate (Sigma-Aldrich, St. Louis, MO) and 4 nM human insulin (Humulin R, Eli Lilly, Indianapolis, IN). Cells were differentiated for 7 days and media was changed every other day.

## Gene silencing

### Adenoviral transduction

On day 3 of differentiation, adipocytes were incubated with 1.5 μl/ml of adenoviral preparation ($10^{12}$ particles/ml) for 24 h in complete culture media containing 1 μg/ml polybrene (hexadimethrine bromide, Sigma-Aldrich, St. Louis, MO). Respirometry, metabolomics, and gene expression were measured on day 7 of differentiation. Ad-mSLC25A11 and Ad-mKate2 shControl were generated by and purchased from Welgen (Worcester, MA).

### siRNA transfection

Undifferentiated pre-adipocytes were transfected with scramble RNA, Mpc1 siRNA, or Aralar1 (Slc25a12) siRNA using Lipofectamine 3000 reagent (Thermo Fisher Scientific, Roskilde, Denmark) according to manufacturer's protocol. In brief, culture media was removed from cells, and cells were incubated with Opti-MEM media (Thermo Fisher Scientific, Roskilde, Denmark), Lipofectamine 3000 reagent and 100 nM scramble RNA or siRNA for 4 h. Then DMEM with 1% fetal bovine serum (Thermo Fisher Scientific, Roskilde, Denmark) was added to the cells and incubated overnight. The next day media was replaced with differentiation media. Respirometry, gene expression and protein expression were measured on day 7 of differentiation. The following siRNAs were used: ON-

TARGETplus Non-targeting Pool (D-001810-10-05), Mouse Mpc1 (55951) siRNA (L-040908-01-0005), and ON-TARGETplus Mouse Slc25a12 siRNA (L-064268-01-0005) from Dharmacon (Lafayette, CO).

### Respirometry measurements

#### Respirometry in intact cells

Pre-treatments were performed in brown adipocyte culture media. The following compounds were used for pre-treatments: 50 nM - 20 μM UK5099 (Sigma-Aldrich, St. Louis, MO), 40 μM Atglistatin (Selleck Chemicals, Houston, TX), 200 μM – 1 mM aminooxyacetic acid (Sigma-Aldrich, St. Louis, MO), 5 μM Triacsin C (Sigma-Aldrich, St. Louis, MO), PF-06424439 (Sigma-Aldrich, St. Louis, MO). The JNJ-DGAT1-A inhibitor (Qi *et al*, 2010) was a generous gift from Janssen Research and Development LLC (Spring House, PA) and is available from MedKoo Biosciences Inc. Prior to respirometry measurements, culture media was replaced with respirometry media (Seahorse XF Base medium, Agilent Technologies, Santa Clara, CA) supplemented with 5 mM glucose and 3 mM glutamine and incubated for 30–45 min at 37°C (without $CO_2$). Where indicated, respirometry media was supplemented with 0.1% fatty acid-free bovine serum albumin (Sigma-Aldrich, St. Louis, MO). During this incubation period, the ports of the Seahorse cartridge were loaded with the compounds to be injected during the assay (50 μl/port) and the cartridge was calibrated. Oxygen consumption rates were measured using the Seahorse XF24-3 extracellular flux analyzer (Agilent Technologies, Santa Clara, CA). The following compounds were used for injections during the assay: 1 μM norepinephrine (Levophed), 4 μM oligomycin A (Calbiochem, San Diego, CA), 40 μM etomoxir (Sigma-Aldrich, St. Louis, MO), 4 μM antimycin A (Sigma-Aldrich, St. Louis, MO), 2 μM Carbonyl cyanide 4-(trifluoromethoxy)phenylhydrazone (FCCP; Sigma-Aldrich, St. Louis, MO). After the assay, cells were fixed using 4% paraformaldehyde (Thermo Fisher Scientific, Roskilde, Denmark). To normalize the data for possible differences in cell number, nuclei were stained with 1 μg/ml Hoechst 33342 (Thermo Fisher Scientific, Roskilde, Denmark) and nuclei were counted using the Operetta High-Content Imaging System (PerkinElmer, Waltham, MA).

#### Respirometry in permeabilized cells

Experiments were performed as previously described in detail (Mahdaviani *et al*, 2015). In brief, differentiated primary brown adipocytes were permeabilized using 5 nM XF PMP reagent (Agilent Technologies, Santa Clara, CA). Respirometry assay was performed in MAS buffer (660 mM mannitol, 210 mM sucrose, 30 mM $KH_2PO_4$, $MgCl_2$, HEPES, EGTA, 1 mM GDP, 1% (w/v) fatty acid-free BSA). The following substrates were used: 5 mM pyruvate, 0.5 mM or 3 mM malate, 5 mM succinate, 2 μM rotenone, 0.1 mM palmitoyl-CoA, 0.5 mM carnitine, 0.1 mM palmitoyl-carnitine, 5 mM ADP. The following compounds were injected: 5 μM oligomycin, 8 μM antimycin a, 0.5 mM N,N,N′,N′-Tetramethyl-p-phenylenediamine (TMPD), 1 mM ascorbic acid, 50 mM sodium azide. Oxygen consumption rates were measured using the Seahorse XF24-3 extracellular flux analyzer (Agilent Technologies, Santa Clara, CA).

### Quantitative real-time PCR

Samples from cells transduced with adenovirus or transfected with siRNA were collected on day 7 of differentiation. RNA was extracted

using the Direct-zol RNA Miniprep Plus Kit® (Zymo Research, Irvine, CA) following the manufacturer's instructions. A sample corresponding to 1 μg RNA from each sample was used to perform cDNA synthesis by the High-Capacity cDNA Reverse Transcription Kit® (Applied Biosystems, Foster City, CA). qPCR was performed using 0.4 ng/μl cDNA and 240 nM of each primer, whose sequences are listed in Table 1.

### Western blot

Protein was isolated using RIPA lysis buffer (Santa Cruz Biotechnology, Dallas, TX) containing protease and phosphatase inhibitor cocktail (Thermo Fisher Scientific, Roskilde, Denmark), and protein concentration was determined using a BCA protein assay (Thermo Fisher Scientific, Roskilde, Denmark). Samples of 10–15 μg isolated protein were diluted in NuPAGE LDS Sample Buffer (Thermo Fisher Scientific, Roskilde, Denmark) containing β-mercaptoethanol (Thermo Fisher Scientific, Roskilde, Denmark), and incubated at 95°C for 5 min. Samples were then loaded into 4–12% Bis-Tris precast gels (Thermo Fisher Scientific, Roskilde, Denmark) and electrophoresed, using NuPAGE MES SDS Running Buffer (Thermo Fisher Scientific, Roskilde, Denmark). Proteins were transferred to methanol-activated Immuno-Blot PVDF Membrane (Bio-Rad, Hercules, CA). Blots were incubated overnight with primary antibody diluted in PBST (phosphate buffered saline with 1 mL/L Tween-20/PBS) + 5% BSA (Thermo Fisher Scientific, Roskilde, Denmark) at 4°C. The next day, blots were washed in PBST and incubated with fluorescent secondary antibodies, diluted in PBST + 5% BSA for 1 h at room temperature. Proteins were detected using the following antibodies: anti-MPC1 (BRP44L Polyclonal Antibody, Thermo Fisher Scientific, Roskilde, Denmark), anti-MPC2 (MPC2 (D4I7G) Rabbit mAb, Cell Signaling, Danvers,

**Table 1.   Primers for qPCR**

| Gene | Forward primer 5'–3' | Reverse Primer 5'–3' |
|------|----------------------|----------------------|
| Mpc1 | GAC TTT CGC CCT CTG TTG CTA | GAG GTT GTA CCT TGT AGG CAA AT |
| Mpc2 | CCG CCG CGA TGG CAG CTG | GCT AGT CCA GCA CAC ACC AAT CC |
| SCL25A11 | AGT CTC CTC TTG GGT GTT AGA | CTT CTG CTT TCT CCT GTC TCC |
| SCL25A12 | TGGTTACCTACGAGCTTCTGC | ACCGATGTGATCGGGGTTG |
| Ucp1 | GGC CTC TAC GAC TCA GTC CA | TAA GCC GGC TGA GAT CTT GT |
| Atgl | TCC GAG AGA TGT GCA AAC AG | CTC CAG CGG CAG AGT ATA GG |
| Pgc1a | AAG ATC AAG GTC CCC AGG CAG TAG | TGT CCG CGT TGT GTC AGG TC |
| Tfam | CAC CCA GAT GCA AAA CTT TCA | CTG CTC TTT ATA CTT GCT CAC AG |
| Elov13 | TCC GCG TTC TCA TGT AGG TCT | GGA CCT GAT GCA ACC CTA TGA |
| Prdm16 | GCC ATG TGT CAG ATC AAC GA | CCT TCT TTC ACA TGC ACC AA |
| 36b4 | GTC ACT GTG CCA GCT CAG AA | TCA ATG GTG CCT CTG GAG AT |

MA), anti-β-Actin (Abcam, Cambridge, United Kingdom), anti-UCP1 (ab10983, Abcam, Cambridge, United Kingdom), anti-Vinculin (V9131, Sigma-Aldrich, St. Louis, MO), Goat anti-Rabbit IgG secondary antibody DyLight 800 (Thermo Fisher Scientific, Roskilde, Denmark). Blots were imaged on the ChemiDoc MP imaging system (Bio-Rad Laboratories, Hercules, CA). Band densitometry was quantified using FIJI (ImageJ, NIH).

**Fluorescence confocal microscopy**

Super-resolution live-cell imaging was performed on a Zeiss LSM880 using a 63× Plan-Apochromat oil-immersion lens and AiryScan super-resolution detector with humidified 5% $CO_2$ chamber on a temperature controlled stage (37°C). Cells were differentiated in glass-bottom confocal plates (Greiner Bio-One, Kremsmünster, Austria). For Lipid droplet imaging cells were incubated with 1 μM Bodipy 558/568 C12 (Thermo Fisher Scientific, Roskilde, Denmark) overnight. The day of imaging cells were stained with 200 nM MitoTracker green (MTG) (Thermo Fisher Scientific, Roskilde, Denmark) for 1 h. MTG and Bodipy 558/568 C12 were removed before imaging and cells were imaged in regular culture media. MTG was excited with 488 nm laser and Bodipy 558/568 C12 was excited with 561 nm laser. For membrane potential measurements cells were incubated with 200 nM MTG and 15 nM Tetramethylrhodamine Ethyl Ester Perchlorate (TMRE) (Thermo Fisher Scientific, Roskilde, Denmark) for 1 h. MTG was washed out prior to imaging and cells were imaged in presence of TMRE in regular culture media. Image Analysis was performed in FIJI (ImageJ, NIH). Image contrast and brightness were not altered in any quantitative image analysis protocols. Brightness and contrast were equivalently modified in the different groups compared, to allow proper representative visualization of the effects revealed by unbiased quantitation.

**Thin-layer chromatography**

Cells were seeded and differentiated in 6-well plate. On day 7 of differentiation, cells were incubated with 1 μM Bodipy 558/568 C12 and DMSO, 100 nM UK5099 or 1 μM norepinephrine for 4 and 24 h. Thin-layer chromatography of intracellular lipids and extracellular lipids was performed as previously described (Rambold *et al*, 2015) with minor modifications. Intracellular lipids and lipids from media were extracted in 500 μl chloroform. Chloroform was evaporated using the Genevac EZ-2 Plus Evaporating System (Genevac, Ipswich, United Kingdom). Lipids were then dissolved in 15 μl chloroform, and 1 μl was spotted on a TLC plate (Silica gel on TLC Al foils, Sigma-Aldrich, St. Louis, MO). Lipids were resolved based on polarity in a developer solution containing ethylacetate and cyclohexane in a 2.5:1 ratio. TLC plates were imaged on the ChemiDoc MP imaging system (Bio-Rad Laboratories, Hercules, CA). Band densitometry was quantified using FIJI (ImageJ, NIH).

**Metabolite tracing**

For metabolomics and stable isotope tracing, cells were cultured and differentiated in 6-well plates using BAT differentiation media. On day 7 of differentiation cells were washed once and treated with 10 μM UK5099 or DMSO for 24 h in DMEM supplemented with phenol red, 10 mM glucose, 2 mM glutamine, and 10% NCS. For palmitate tracing experiments NCS was delipidated using fumed silica (Sigma-Aldrich, St. Louis, MO), and 200 μM [U-$^{13}$C$_{16}$] palmitate (Cambridge Isotope Laboratories, Tewksbury, MA) was added to the media, with unlabeled palmitate added to matched controls. Palmitate was added at a 4:1 palmitate:BSA complex. For glutamine-tracing experiments, unlabeled glutamine in the media was replaced with 2 mM [U-$^{13}$C$_5$] glutamine (Cambridge Isotope Laboratories, Tewksbury, MA). Metabolite extraction and GC/MS was performed as previously described in detail (Vacanti *et al*, 2014).

**G3P and DHAP measurements**

G3P and DHAP were measured using fluorimetric and colorimetric kits available from Sigma-Aldrich (MAK207 and MAK275, respectively). After differentiation in 6-well plates, cells were treated with DMSO (vehicle), 100 nM UK5099 or 1 μM Norepinephrine for 24 h. Then, cells were washed with PBS and 60 μl of the respective assay buffer were added to each well. The suspension was centrifuged for 2,000 × *g* for 1 min, and 10 μl of supernatant were used to measure G3P and DHAP according to the manufacturer instructions. For normalization purposes, 2 μl were used to measure protein concentration by a BCA protein assay kit (Thermo Fisher Scientific, Roskilde, Denmark).

**Statistical analyses**

Data were presented as mean ± SEM for all conditions. Comparisons between groups were done by one-way ANOVA with Tukey's or Holm Sidak's test for pairwise multiple comparisons. When appropriate, two-way ANOVA with Tukey's multiple comparisons test was employed. Pairwise comparisons were done by two-tailed Student's *t*-test. Differences of $P < 0.05$ were considered to be significant. All graphs and statistical analyses were performed using GraphPad Prism 8 for Windows (GraphPad Software, San Diego, CA).

# Data availability

This study contain no data that were deposited in a public database.

**Expanded View** for this article is available online.

## Acknowledgements

We would like to thank, Dr. Nathanael Miller for assistance with image analysis. We would like to thank Dr. Dani Dagan, Dr. Evan Taddeo, Dr. Karel Erion, Dr. Michael Shum, and Jennifer Ngo for helpful discussions and advice. O.S.S. is funded by NIH-NIDDK 5-RO1DK099618-02. M.F.O. is funded by Conselho Nacional de Desenvolvimento Científico e Tecnológico (grants #229526/2013-6, #404153/2016-0 and #303044/2017-9) the Fundação Carlos Chagas Filho de Amparo à Pesquisa do Estado do Rio de Janeiro (FAPERJ) (grants #E-26/102.333/2013 and #E-26/203.043/2016) and by the Coordenação de Aperfeiçoamento de Pessoal de Nível Superior–Brasil (CAPES)–Finance Code 001 and CAPES-PrInt. M.L. is funded by the Department of Medicine chair commitment at UCLA, Pilot and Feasibility grants from NCATS UL1TR001881 (CTSI), NIDDK P30 DK063491 (UCSD-UCLA DERC), NIDDK P30 41301 (CURE:Digestive Diseases Research Center) and 1R01AA026914-01A1. A.S.D. is funded by NIH R35GM138003. A.E.J. is funded by the UCLA Tumor Cell Biology Training

Program (USHHS Ruth L. Kirschstein Institutional National Research Service Award #T32 CA009056). E.A.A. is funded by the Azrieli and Foulkes foundations. Special thanks to Barbara Cannon and Jan Nedergaard for guiding our labs into the field of thermogenesis and brown adipocyte biology.

## Author contributions

Conceptualization; MV, ML, MFO, OSS, Investigation; MV, CMF, IYB, AEJ, AB, BRD, KM, RA-P, AP, EAA, LS, MFO, Writing-Original Draft; MV, MFO, ML, OSS. Writing-Review & Editing; MV, ASD, MP, BEC, ML, MFO, OSS.

## Conflict of interest

A.S.D. has previously served as a paid consultant for Agilent Technologies.

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
