## [Review Process File · EMBO Reports]

Blocking mitochondrial pyruvate import in brown adipocytes induces energy wasting via lipid cycling

Michaela Veliova, Caroline Mendes Ferreira, Ilan Benador, Anthony Jones, Kiana Mahdavian, Alexandra Brownstein, Brandon Desousa, Rebeca Acín-Pérez, Anton Petcherski, Essam Assali, Linsey Stiles, Ajit Divakaruni, Marc Prentki, Barbara Corkey, Marc Liesa, Marcus Oliveira, and Orian Shirihai

DOI: 10.15252/embr.201949634

Corresponding author(s): Orian Shirihai (oshirihai@mednet.ucla.edu), Marc Liesa (mliesa@mednet.ucla.edu), Marcus Oliveira (marcusoliveiraufjr@gmail.com)

Review Timeline:

Submission Date:	12th Nov 19
Editorial Decision:	10th Dec 19
Revision Received:	10th Jul 20
Editorial Decision:	17th Aug 20
Revision Received:	15th Sep 20
Accepted:	7th Oct 20

Editor: Deniz Senyilmaz Tiebe

Transaction Report:

Dear Orian,

Thank you for submitting your manuscript, which was now seen by two referees, whose reports are copied below.

As you can see, the referees express interest in the proposed role of MPC in brown adipose tissue function. However, they also raise a number of concerns that need to be addressed to consider publication here. I find the reports informed and constructive, and believe that addressing the concerns raised will significantly strengthen the manuscript.

Given these constructive comments, we would like to invite you to revise your manuscript with the understanding that the referee concerns (as in their reports) must be fully addressed and their suggestions taken on board. Please address all referee concerns in a complete point-by-point response. Acceptance of the manuscript will depend on a positive outcome of a second round of review. It is EMBO reports policy to allow a single round of revision only and acceptance or rejection of the manuscript will therefore depend on the completeness of your responses included in the next, final version of the manuscript.

1. A data availability section providing access to data deposited in public databases is missing (where applicable).
2. Your manuscript contains statistics and error bars based on $n=2$ or on technical replicates. Please use scatter plots in these cases.

Supplementary/additional data: The Expanded View format, which will be displayed in the main HTML of the paper in a collapsible format, has replaced the Supplementary information. You can submit up to 5 images as Expanded View. Please follow the nomenclature Figure EV1, Figure EV2 etc. The figure legend for these should be included in the main manuscript document file in a section called Expanded View Figure Legends after the main Figure Legends section. Additional Supplementary material should be supplied as a single pdf labeled Appendix. The Appendix includes a table of content on the first page with page numbers, all figures and their legends. Please follow the nomenclature Appendix Figure Sx throughout the text and also label the figures according to this nomenclature. For more details please refer to our guide to authors.

2) individual production quality figure files as .eps, .tif, .jpg (one file per figure).

3) a .docx formatted letter INCLUDING the reviewers' reports and your detailed point-by-point responses to their comments. As part of the EMBO Press transparent editorial process, the point-by-point response is part of the Review Process File (RPF), which will be published alongside your paper. For more details on our Transparent Editorial Process, please visit our website:

<https://www.embopress.org/page/journal/14693178/authorguide#transparentprocess>

4) a complete author checklist, which you can download from our author guidelines (<http://embor.embopress.org/authorguide>). Please insert information in the checklist that is also reflected in the manuscript. The completed author checklist will also be part of the RPF.

5) Please note that all corresponding authors are required to supply an ORCID ID for their name upon submission of a revised manuscript (<https://orcid.org/>). Please find instructions on how to link your ORCID ID to your account in our manuscript tracking system in our Author guidelines (<http://embor.embopress.org/authorguide>).

6) We replaced Supplementary Information with Expanded View (EV) Figures and Tables that are collapsible/expandable online. A maximum of 5 EV Figures can be typeset. EV Figures should be cited as 'Figure EV1, Figure EV2' etc... in the text and their respective legends should be included in the main text after the legends of regular figures.

- For the figures that you do NOT wish to display as Expanded View figures, they should be bundled together with their legends in a single PDF file called *Appendix*, which should start with a short Table of Content. Appendix figures should be referred to in the main text as: "Appendix Figure S1, Appendix Figure S2" etc. See detailed instructions regarding expanded view here: <http://embor.embopress.org/authorguide#expandedview>.

7) We would also encourage you to include the source data for figure panels that show essential data.

Numerical data should be provided as individual .xls or .csv files (including a tab describing the data). For blots or microscopy, uncropped images should be submitted (using a zip archive if multiple images need to be supplied for one panel). Additional information on source data and instruction on how to label the files are available <http://embor.embopress.org/authorguide#sourcedata>.

8) Regarding data quantification, please ensure to specify the name of the statistical test used to generate error bars and P values, the number (n) of independent experiments underlying each data point (not replicate measures of one sample), and the test used to calculate p-values in each figure legend. Discussion of statistical methodology can be reported in the materials and methods section,

but figure legends should contain a basic description of n, P and the test applied.
Please note that error bars and statistical comparisons may only be applied to data obtained from at least three independent biological replicates.
Please also include scale bars in all microscopy images.

I look forward to seeing a revised version of your manuscript when it is ready. Please let me know if you have questions or comments regarding the revision.

Kind regards,

Deniz

Deniz Senyilmaz Tiebe, PhD
Editor
EMBO Reports

Referee #1:

Veliova and colleagues examined the role of the mitochondrial pyruvate carrier (MPC) in brown adipose tissue and discovered a mitochondrial fuel-dependent mechanism regulating futile energy expenditure. Specifically, MPC activity was targeted in primary brown adipocytes using pharmacologic and genetic means to examine oxidative rates under basal and stimulated conditions. They found glutamine-driven and malate-aspartate shuttle-supported fatty acid oxidation increases with loss of MPC activity in BAT. Additional studies demonstrated fatty acid oxidation was increased to maintain ATP levels required to convert free fatty acids and triglyceride-derived fatty acids to acyl-CoA primed for oxidation. This futile cycle was proposed to selectively increase energy expenditure via specific BAT targeting of the MPC. This reviewer appreciates the effort to use inhibitors such as etomoxir at concentrations to limit off-target effects.

Major weaknesses:

- 1) Characterization malate-aspartate shuttle involvement to supply oxaloacetate for fatty acid oxidation feels incomplete. Why was only the OGC investigated and not the Aralar proteins?
- 2) If OGC is supplying malate to the TCA cycle for oxaloacetate production independent of Aralar, where is the required aKG for antiport coming from? Does this explain the need for glutamine?
- 3) Aminooxyacetic acid is described as have specific effect on the malate-aspartate shuttle, but AOA would likely affect all transaminase activities including that of alanine-transaminases known to serve as a bypass of MPC blockade. Moreover, AOA at concentrations below what was used in this study is shown to form adducts with other carboxylates which may confound the interpretation of these data (Yang L, et al. 2008. J Biol Chem. 283: 21978-87. PMID: 18544527).

Minor weaknesses:

- 1) Figure S2 does not exist as a reference in the text or as an actual figure

2) Figure legends do not always match a given figure panel. i.e. Figure 3B which shows glucose+glutamine and glucose OCR of non-stimulated, NE-stimulated, and Eto sensitive respiration. The legend reads "Quantification of basal and etomoxir-sensitive OCR after vehicle or UK5099 treatment." UK5099 is in no way denoted in the figure.

3) Why was FBS used in figure 1E and not palmitate conjugated BSA? FBS contains many things that could affect mitochondria/cellular metabolism in addition to fatty acid conjugated BSA.

4) Why are vehicle treatments shown with error bars in some figure panels but not others?

Referee #2:

In this paper, Veliova et al. report that inhibition of the mitochondrial pyruvate carrier (MPC) in brown adipocytes leads to increased oxygen consumption and OXPHOS-linked ATP production through increased fatty oxidation. Importantly, increased fatty oxidation does not lead to mitochondrial uncoupling, even when adipocytes are stimulated with NE. This increased ATP production appears to fuel futile lipid cycling resulting in re-esterification of fatty acids into triglycerides. Inhibition of this futile cycle reduces the increased oxygen consumption triggered by MPC inhibition. Interestingly, inhibition of MPC bypasses the need for adrenergic stimulation of mitochondrial uncoupling. Although the results are sound, novel and interesting, there are a few points that should be clarified further.

1) It has been previously shown that Thiazolidinedione drugs can promote glyceroneogenesis and in particular stimulate glycerol kinase activity. Given that in addition to acting as PPAR gamma agonists, these compounds are known to inhibit the MPC, it would be interesting to investigate whether UK5099 can also affect this pathway.

2) I understand that GL/FFA cycling can be a composite of many shorter cycles. But in this case, why would inhibition of ATGL with Atglistatin or inhibition of the acetyl-CoA synthase with Triacsin C be sufficient to blunt the effect of MPC inhibition on OCR?

3) What is the level of UCP1 in the primary cultures used in this study? Could a low level explain why increased fatty acid oxidation is not accompanied by mitochondrial proton leakage?

4) Knock down of MPC1 is only partial (Figure 3C), which may explain the lack of NE-stimulated OCR compared to UK5099 (Figure 3F,G). This could be discussed. Have the authors tested additional MPC inhibitors, including Thiazolidinediones?

5) Do the inhibitors used in Figure 4 have an effect on OCR in untreated cells?

Referee #1:

Veliova and colleagues examined the role of the mitochondrial pyruvate carrier (MPC) in brown adipose tissue and discovered a mitochondrial fuel-dependent mechanism regulating futile energy expenditure. Specifically, MPC activity was targeted in primary brown adipocytes using pharmacologic and genetic means to examine oxidative rates under basal and stimulated conditions. They found glutamine-driven and malate-aspartate shuttle-supported fatty acid oxidation increases with loss of MPC activity in BAT. Additional studies demonstrated fatty acid oxidation was increased to maintain ATP levels required to convert free fatty acids and triglyceride-derived fatty acids to acyl-CoA primed for oxidation. This futile cycle was proposed to selectively increase energy expenditure via specific BAT targeting of the MPC. This reviewer appreciates the effort to use inhibitors such as etomoxir at concentrations to limit off-target effects.

Major weaknesses:

1) Characterization malate-aspartate shuttle involvement to supply oxaloacetate for fatty acid oxidation feels incomplete. Why was only the OGC investigated and not the Aralar proteins?

Answer 1) We thank this referee for bringing up this important point and agree that assessing the requirement for Aralar will make our study more complete. To address this point we have added new data to the manuscript, in which we determined the role of Aralar in the increased energy demand that is induced by blocking the MPC. We chose to silence SLC25A12 (Aralar1), which is the predominant isoform of Aralar in brown adipose tissue. Our new data in Figure 3 shows that siRNA mediated knock down of Aralar reverses the effects of MPC inhibition on non-stimulated and etomoxir-sensitive respiration similarly to OGC1 knock-down. This new data is supporting our hypothesis, that the malate aspartate shuttle activity is required to support fat oxidation, energy expenditure and glutamine metabolism when MPC is inhibited.

New data added:**Figure 4F-I****New text (page9):**

The involvement of the MASH in UK5099-induced energy expenditure was further assessed by silencing another MASH component, the mitochondrial aspartate/glutamate carrier (SLC25A12 or Aralar1). Aralar1 catalyzes the calcium-dependent exchange of aspartate and glutamate

through the inner mitochondrial membrane, carrying glutamate inside mitochondria [36]. Aralar is found as two isoforms 1 and 2 (Aralar 1 or SLC25A12, and Aralar 2 or citrin or SLC25A13), with Aralar1 being the most abundant isoform in BAT [37,38]. siRNA-mediated knock-down of Aralar was confirmed by qPCR in primary brown adipocytes (Fig EV4D). Similar to OGC1 KD cells, Aralar1 KD reversed the stimulatory effects of MPC inhibition on respiratory rates in non-stimulated and NE-stimulated brown adipocytes (Figs 4F-G, EV4E). Furthermore, knock-down of Aralar1 reversed the effects of MPC inhibition on NE-stimulated mitochondrial fat oxidation as revealed by the lack of an increase in etomoxir-sensitive respiration following UK5099 treatment (Fig 4H).

Next, we sought to determine whether the MASH is required for the increase in glutamine metabolism. To test this hypothesis, we analyzed the effect of UK5099 treatment on glutamate and aspartate abundance in Aralar1 KD as compared to scrambled siRNA treated brown adipocytes. Here we applied the same principle as in Figure 3D where the ratio of aspartate to glutamate is used as a measure of glutamine catabolism. Knock-down of Aralar1 resulted in partial reversal of the increase in aspartate to glutamate ratio induced by MPC inhibition (Fig 4I). Partial reversal was expected, given that glutamine can provide glutamate and alpha-ketoglutarate independently of the MASH, through mitochondrial glutaminase and glutamate dehydrogenase. Thus, our data indicate that the Malate-Aspartate Shuttle is required to allow an increase in Energy Expenditure in Brown Adipocytes (MASH_{EEBA}) via glutamate and fatty acid oxidation under MPC inhibition. MASH_{EEBA} might provide the extra electrons needed to synthesize the amount of ATP required to cover the increase in ATP demand induced by MPC inhibition. Thus, we next asked the question which process was activated by MPC inhibition to increase ATP demand and consequently respiration.

2) If OGC is supplying malate to the TCA cycle for oxaloacetate production independent of Aralar, where is the required aKG for antiport coming from? Does this explain the need for glutamine?

Answer 2) We thank the referee for bringing up this interesting point. In the revised manuscript we added the following sets of data that addresses this point: **1.** Our new data using siRNA for Aralar indicates that Aralar activity is required for UK5099-induced OCR. **2.** We added new metabolomics data, that provides the aspartate to glutamate ratio as a measure of glutaminolysis and aminotransferase activity. This experiment shows that in control cells, UK5099 increases aspartate to glutamate ratio. This effect of UK5099 is diminished when Aralar is knocked-down, further supporting that Aralar activity is required for glutamine oxidation. **3.** To more specifically determine the contribution of glutamine to TCA metabolites following MPC inhibition, we performed ¹³C glutamine tracing in cells treated with UK5099 or vehicle. Our new data shows that the incorporation of glutamine derived carbons to TCA cycle metabolites is increased upon UK5099 treatment. When we analyzed the isotopomer distribution of ¹³C glutamine, we found that the isotopomer that was most affected by UK5099 treatment was the M+3 isotopomer, which is indicative of reductive carboxylation of α-ketoglutarate to citrate. These data indicate that glutamine derived carbons are utilized for *de novo* lipogenesis and thereby provide a mechanism for brown adipocytes to sustain lipid synthesis when mitochondrial pyruvate import is limited.

New data
Figure 3D:

New text (page 8):

Glutaminase generates glutamate in the cytosol and mitochondria. Glutamate can be oxidized to alpha-ketoglutarate and enter the TCA cycle. Moreover, glutamate and oxaloacetate can be transaminated to alpha-ketoglutarate and aspartate in the mitochondria and in the cytosol. As a consequence, the ratio of aspartate/glutamate can be used as a measure of glutamine catabolism. We thus reasoned that, if UK5099 increased glutamate entry to the TCA cycle, glutamate abundance should decrease, while aspartate should increase, and their ratio would increase [16,19]. We found that UK5099 treatment increased aspartate to glutamate ratio, indicating increased glutamine catabolism to glutamate and transaminase activity to generate aspartate (Fig 3D).

Figure EV3

New text (page 8):

To specifically determine the metabolic fate of glutamine into the TCA cycle intermediates, we traced the incorporation of glutamine carbons to polar metabolites, using uniformly labeled glutamine [U-13C5] and quantifying labeled metabolite enrichment by GC-MS (Fig EV3B). UK5099 significantly increased the incorporation of glutamine-derived carbons into aspartate,

glutamate, α -ketoglutarate and malate, thereby supporting the hypothesis that glutamine is required for TCA anaplerosis, when mitochondrial pyruvate import is limited (Fig EV3C). Interestingly, the isotopomer distribution shows, that most of the increase in glutamine derived carbons were in the M+3 mass isotopomer (Fig EV3D), which is indicative of reductive carboxylation of glutamine and increased alpha-ketoglutarate availability [31].

3) Aminoxyacetic acid is described as have specific effect on the malate-aspartate shuttle, but AOA would likely affect all transaminase activities including that of alanine-transaminases known to serve as a bypass of MPC blockade. Moreover, AOA at concentrations below what was used in this study is shown to form adducts with other carboxylates which may confound the interpretation of these data (Yang L, et al. 2008. J Biol Chem. 283: 21978-87. PMID: 18544527).

Answer 3) We thank the referee for bringing up this important point and we agree that the use of AOA for the inhibition of malate aspartate shuttle activity has its limitations. To that extent we have we have now pointed out potential off-targets of AOA in the revised version.

To address whether the effects of AOA targeting other transaminases were contributing to the phenotype, we tested whether specific knock-down of malate-aspartate shuttle components were recapitulating the results observed with AOA. To test this we performed: **1.** shRNA mediated knock-down of OGC1. **2.** siRNA mediated knock-down of Aralar1. As the genetic approaches recapitulated the pharmacological inhibition of malate-aspartate shuttle using AOA, we are further validating the involvement of the malate-aspartate shuttle in supporting increased respiration upon MPC inhibition. These data are our added to the revised version.

New text added (page 9):

To further confirm the involvement of the MASH in the metabolic effects caused by UK5099, and address potential off-target effects of AOA [33,34], we then silenced the expression of oxoglutarate carrier 1 (SLC25A11 or OGC1), a key component of MASH.

Minor weaknesses:

1) Figure S2 does not exist as a reference in the text or as an actual figure

Answer 1) We thank the referee for noticing this mistake. We now have fixed all figure numbers and crosschecked that they are correctly referenced in the text.

2) Figure legends to not always match a given figure panel. i.e. Figure 3B which shows glucose+glutamine and glucose OCR of non-stimulated, NE-stimulated, and Eto sensitive respiration. The legend reads "Quantification of basal and etomoxir-sensitive OCR after vehicle or UK5099 treatment." UK5099 is in no way denoted in the figure.

Answer 2) We thank the referee for noticing our mistakes. All figure legends were updated.

3) Why was FBS used in figure 1E and not palmitate conjugated BSA? FBS contains many things that could affect mitochondria/cellular metabolism in addition to fatty acid conjugated BSA.

Answer 4) We agree that respirometry data utilizing FBS could be difficult to interpret. We therefore removed FBS data from the manuscript and focus solely on the effects of fatty acid free BSA.

4) Why are vehicle treatments shown with error bars in some figure panels but not others?

Answer 5) Panels showing vehicle data in absolute values also included error bars. However, in some experiments data is presented relative to vehicle where the y-axis is "fold change over vehicle". Because efficiency of brown adipocyte differentiation in culture varies in independent

experiments between 70-90%, and OCR is highly sensitive to differentiation efficiency, this variability can cause big differences in the absolute OCR. Thus to minimize this variability and truly assess the effects of MPC inhibition, we represent the data as fold change over vehicle for each independent experiment. OCR are much more sensitive to variability in differentiation efficiency than other experiments, therefore experiments other than respirometry measurements were represented as absolute values.

Referee #2:

In this paper, Veliova et al. report that inhibition of the mitochondrial pyruvate carrier (MPC) in brown adipocytes leads to increased oxygen consumption and OXPHOS-linked ATP production through increased fatty oxidation. Importantly, increased fatty oxidation does not lead to mitochondrial uncoupling, even when adipocytes are stimulated with NE. This increased ATP production appears to fuel futile lipid cycling resulting in re-esterification of fatty acids into triglycerides. Inhibition of this futile cycle reduces the increased oxygen consumption triggered by MPC inhibition. Interestingly, inhibition of MPC bypasses the need for adrenergic stimulation of mitochondrial uncoupling. Although the results are sound, novel and interesting, there are a few points that should be clarified further.

1) It has been previously shown that Thiazolidinedione drugs can promote glyceroneogenesis and in particular stimulate glycerol kinase activity. Given that in addition to acting as PPAR gamma agonists, these compounds are known to inhibit the MPC, it would be interesting to investigate whether UK5099 can also affect this pathway.

Answer 1) This is indeed a very interesting point. In fact, our data (Figs 5E and EV5E) show increased G3P levels treatment pointing towards increased glycerol kinase activity or increased glycerol 3 phosphate dehydrogenase activity. Since TZDs increased glycerol kinase by activating PPARg and activating glycerol kinase transcription we expect this mechanism would be independent of MPC blockage. Therefore, we do not expect that UK5099 increases glycerol kinase activity through the same mechanism described for TZDs. Indeed, we expect that higher G3P levels upon MPC inhibition reflect an increased flux of metabolites towards lipogenesis. We have added these important points in the revised version of the discussion

New text added (page 14):

Importantly, FDA-approved drugs, such as Thiazolidinediones (TZDs), were shown to target the MPC at clinically relevant concentrations [59]. Remarkably, TZDs were shown to promote glyceroneogenesis and lipid cycling through increased expression of glycerol kinase (GK) in human adipose tissue [60]. However, this effect was a result of PPARgamma mediated activation of GK transcription, and therefore can likely not be attributed to TZDs effect on the MPC. Further work is required to assess the contribution of lipid cycling and the MASH in the beneficial effects of TZDs.

2) I understand that GL/FFA cycling can be a composite of many shorter cycles. But in this case, why would inhibition of ATGL with Atglistatin or inhibition of the acetyl-CoA synthase with Triacsin C be sufficient to blunt the effect of MPC inhibition on OCR?

Answer 2) We thank the referee for raising this question and giving us the opportunity to clarify this point. Indeed, we show that, both inhibition of ATGL and AcylCoA Synthetase (ACS) were sufficient to reverse UK5099 effects, whereas blocking DGAT enzymes had no effect on UK5099 induced ATP demand. These results suggest that MPC inhibition does not involve the esterification of fatty acids into TAGs through a DGAT dependent pathway, but rather through a sub-cycle to DAGs or MAGs in a DGAT-independent pathway. The data showing the ATGL inhibition fully reversed UK5099 effects might suggest that energy demand created by MPC inhibition mostly involves fatty acid esterification from the endogenous TAG stores. Overall, the

data indicate that the cycles are not equally contributing and equally available to act as an alternative to each other and that flux through ATGL and ACS have a higher ATP demand and are essential in maintaining lipid cycling whereas flux through DGAT enzymes is not. We have added a paragraph in the revised version discussing this important point.

To further support that, upon MPC inhibition, ATP demand is the main driver of increased OCR, we added new respirometry data in permeabilized cells with pyruvate, malate, palmitoyl CoA and carnitine as substrates. The advantage of using permeabilized cells to address this question is that ATP demand in this system is dictated by the amount of ADP supplied in the assay buffer. Therefore, in this system an increase in OCR reflects an increase in respiration that is independent of ATP demand. Our new data shows that given both pyruvate, malate and fatty acids as substrates, MPC inhibition does not change OCR, thus suggesting that the increase in OCR in the intact cells is mainly driven by an increase in ATP demand.

New data (Figure 5C):

New Text (page 10):

We reasoned that if ATP demand is elevated by MPC inhibition, then UK5099 treatment should have no effect on mitochondrial respiration when cellular ATP demand is clamped, by controlling cytosolic ADP concentrations. To test this hypothesis, we clamped the cytosolic ADP by permeabilizing brown adipocytes and supplementing the assay buffer with 5 mM ADP. OCR were measured after concurrently providing all fuels (pyruvate, malate, palmitoyl-CoA and carnitine). UK5099 treatment did not change mitochondrial respiration in permeabilized cells when given all substrates and 5 mM ADP (Fig 5C). This observation strengthens that increased mitochondrial respiration in intact brown adipocytes induced by MPC inhibition is mainly driven by increased ATP demand.

New Text (page 12-13):

We show that MPC in brown adipocytes increases energy expenditure through the activation of ATP demanding lipid cycling. Using pharmacological inhibitors of various steps in TAG breakdown or esterification we show that ATGL-dependent lipolysis and ACS-dependent fatty acid esterification are required for the increased energy demand induced by MPC inhibition (Figs 5F-K). Interestingly we find that inhibition of DGAT enzymes had no effect on energy demand under UK5099 treatment (Figs 5H-I). These data suggest that MPC inhibition activates a sub-cycle in the lipid cycling pathway which involves ACS-dependent esterification of lipids. Furthermore, our

data suggest that the majority of the lipids used in the sub-cycle are from endogenous TAG stores that are released from LDs in an ATGL-dependent manner. However, it is conceivable that DGAT inhibition using pharmacological inhibitors was incomplete and therefore had no apparent effect on energy demand following MPC inhibition. It remains to be determined by which mechanism inhibition of the MPC stimulates lipolysis and lipid cycling. In this regard, ROS activate lipolysis in white adipocytes [54] and blocking MPC decreases the synthesis of the ROS-scavenger glutathione, by diverting glutamine away from glutathione biosynthesis [55]. Thus, it is a possibility that lipolysis and the concomitant lipid cycling are initiated by redox-signaling induced by MPC blockage.

3) What is the level of UCP1 in the primary cultures used in this study? Could a low level explain why increased fatty acid oxidation is not accompanied by mitochondrial proton leakage?

Answer 3) We thank the referee for bringing up this important point. Indeed, adrenergically stimulated lipolysis is associated with UCP1 dependent proton leak. However, it was previously shown that fatty acids alone are not enough to induce UCP1 activation in the absence of adrenergic stimulation (DOI 10.1002/embj.201385014). Therefore, we do not expect increased UCP1 activity upon UK5099 induced lipolysis in non-stimulated brown adipocytes. However, we want to point out that MPC inhibition leads to increased ATP linked respiration, under both non-stimulated and NE-stimulated conditions, but this does not happen at the expense of mitochondrial proton leak. To clarify this point we have added quantifications of ATP linked respiration and mitochondrial proton leak to Figures 2 and 4.

To further address this comment and to ensure that UCP1 levels are not affected by UK5099 treatment or MPC1 knock-down we have added new data including Western blot analysis and qPCR analysis probing for UCP1. Our new data show no changes in UCP1 protein or mRNA levels upon UK5099 treatment or MPC1 knock-down. These data are added to Extended View Figure 2.

New data Figure 2 B

New text (page 6):

Sympathetic stimulation of brown adipocytes increase mitochondrial fatty acid oxidation by stimulating lipolysis uncoupling mitochondria. Consequently, acute treatment of brown adipocytes with norepinephrine (NE) increases mitochondrial fat oxidation to produce heat by activating UCP1. Thus, we next aimed to determine whether MPC inhibition would further increase fatty acid oxidation in activated brown adipocytes or whether it would be inert, as expected with MPC1 inhibition in uncoupled and depolarized mitochondria. We find that MPC inhibition by UK5099 further increased OCR in NE-stimulated brown adipocytes (Figs 2A-B).

New Data (Figure 5B):

New Text (page 10)

Intracellular handling of nutrients, particularly of fatty acids, is an ATP demanding process. Our data and others demonstrated that when mitochondria do not have access to pyruvate, a cellular response ensures a switch to use fatty acids as an oxidative fuel in the mitochondria to synthesize ATP. Accordingly, in addition to the increase in mitochondrial fat oxidation induced by MPC inhibition, we find that UK-5099 treatment increases ATP-synthesizing respiration both in NE-stimulated (Fig. 2A, 3B) and non-stimulated brown adipocytes in a dose-dependent manner (Fig 5A-B, EV5C). Further supporting the increase in ATP-synthesizing fat oxidation induced by MPC inhibition, UK5099 treatment induced an increase in mitochondrial membrane potential in non-stimulated brown adipocytes (Figure EV5A-B). Hyperpolarization of mitochondria can explain the increase in oligomycin insensitive respiration induced by UK treatment as well (Fig 5A-B) [39].

New Data (Figure EV2):

New text (page 7)

To address the possibility that UK5099 increased coupled respiration by decreasing UCP1 content, we measured UCP1 expression in cells treated with UK5099. Western blot and qPCR analyses showed that UK5099 treatment did not affect UCP1 expression and protein content (Fig EV2A). As in UK5099 treated brown adipocytes, qPCR analysis confirmed that MPC1 KD did not reduce UCP1 expression or affect brown adipocyte differentiation markers compared to control cells (Fig EV2B).

4) Knock down of MPC1 is only partial (Figure 3C), which may explain the lack of NE-stimulated OCR compared to UK5099 (Figure 3F,G). This could be discussed.

Answer 4) We thank the referee for raising this important point. We have included a discussion of this point in the revised discussion section.

New text (page 13):

Interestingly the effect of knock-down of MPC1, while being similar to effects of UK5099-treatment in the absence of NE, did not reproduce the same effect in NE-stimulated brown adipocytes. The difference between the effects of pharmacological and genetic interference on NE-induced energy expenditure may be attributed to differences in level of MPC inhibition between UK5099 and genetic knock-down of MPC1 or recruitment of compensatory mechanisms in MPC1 knock-down cells that are not induced upon 2 hours of UK5099 treatment.

5) Have the authors tested additional MPC inhibitors, including Thiazolidinediones?

Answer 5) We thank the referee for bringing up the idea of expanding this study towards the effects of TZDs. We agree that this is a natural direction to adopt in our path to translate this study towards clinical application. However, to properly address the contribution of the pathways described in our study to the beneficial effect of TZDs, we decided to run a comprehensive study that will compare the effect of various TZDs with higher and lower ratio of MPC/PPARgamma engagement. We believe these results will better fit in a future publication, as this requires its own set of figures. Beyond TZDs and UK5099, several other MPC inhibitors are now known, including lonidamine (DOI: 10.1042/BJ20151120), tolylfluanid (DOI: 10.1210/en.2017-00695) and zaprinast (DOI: 10.1074/jbc.M113.507285). Zaprinast was used for the development of sildenafil and is a classical inhibitor of cGMP-specific phosphodiesterase (PDE). Interestingly, recent evidence demonstrated that Zaprinast inhibits MPC in concentrations far lower than usually employed to target PDE. Remarkably, in low concentrations Zaprinast increased aspartate/glutamate ratio in an Aralar-independent mechanism. We then decided to test the effects of zaprinast on brown adipocytes respiration. Our preliminary results from two independent experiments demonstrated that Zaprinast treatment (up to 20 μ M) caused no significant effects on non-stimulated brown adipocyte respiration (see figure L1 below). However, upon activation with norepinephrine, zaprinast caused remarkable reductions in respiratory rates at higher (>10 μ M) concentrations. These results indicate that zaprinast might have off-target effects in brown adipocytes beyond MPC inhibition that ultimately affect respiration. Indeed, zaprinast was shown to inhibit glutaminase activity (DOI: 10.1158/2159-8290.CD-13-0572), which might explain the inhibitory effects on respiration in brown adipocytes. Despite zaprinast inhibiting MPC and increasing aspartate/glutamate ratio in other cells types, it does so without the involvement of Aralar. Therefore, we think the response of brown adipocytes to zaprinast might result from mixed effects on targets beyond the MPC. For this reason, we chose not to include these preliminary data.

Figure L1: Effect of zaprinast treatment on brown adipocytes respiratory rates. Cells were treated for 2h in DMEM media + 10% NCS with different zaprinast concentrations and changed to seahorse media containing 3 mM glucose and 3 mM glutamine and different zaprinast concentrations. (A) Effect of 0-5 μ M Zaprinast on OCR in non-activated brown adipocytes. (B) Effect of Zaprinast on OCR in brown adipocytes. Cells were incubated with 0 (control, blue trace), 5 μ M (red trace), 10 μ M (green trace) and 20 μ M Zaprinast (yellow trace). Arrows above represents the injections of norepinephrine (NE), FCCP, etomoxir and antimycin a (AA).

6) Do the inhibitors used in Figure 4 have an effect on OCR in untreated cells?

Answer 6) We thank the referee for bringing up this important point, which we addressed with new data in the revised version. In the revised version, we included new data showing the effect of DGAT inhibitors and Triacsin C treatment on cells where MPC is not inhibited. The data shows that treatment with DGAT inhibitors does not affect non-stimulated and ATP-linked OCR. Treatment with Triacsin C has an inhibitory effect on non-stimulated and ATP-linked OCR. This data agrees with previous studies showing that lipid esterification contributes to basal energy demand in adipose tissue (DOI: 10.1194/jlr.M050005). In addition, the metabolic state under MPC inhibition is a different one than in untreated cells, as MPC inhibition forces the cells to activate lipolysis and lipid cycling therefore the effect of Triacsin C on OCR in vehicle vs UK5099 treated cells could be due to different mechanisms.

New data: Figure 5H-K:

New text (page 11):

To assess the potential role of TAG synthesis in UK5099-mediated activation of lipid cycling, we blocked the last step of TAG synthesis catalyzed by diglyceride acyltransferase 1 and 2 (DGAT1/2) using pharmacological inhibitors (JNJ compound A [43] and PF-06424439). UK5099-induced respiratory rates were insensitive to DGAT inhibition (Figs 5H-I). These results suggest that ATP-demand induced by MPC inhibition does not involve the last step of TAG synthesis. We then blocked acyl-CoA synthetase (ACS) by using Triacsin C, thereby blocking a sub-cycle of lipid cycling (Figure 5D). UK5099-induced increase in ATP-synthesizing respiration was highly sensitive to Triacsin C, reaching values close to vehicle treated cells (Figs 5J-K). In the absence of UK5099, Triacsin C reduced basal and ATP-synthesizing respiration in non-stimulated brown adipocytes, in agreement with previous studies showing that lipid esterification contributes to basal ATP demand in adipose tissue [42].

Dear Orian,

Thank you for submitting the revised version of your manuscript. It has now been seen by both of the original referees.

As you can see, the referees find that the study is significantly improved during revision and recommend publication. Before I can accept the manuscript, I need you to address some minor points below:

- Please address the remaining minor concern of referee #1.
- Please provide 3-5 keywords for your study. These will be visible in the html version of the paper and on PubMed and will help increase the discoverability of your work.
- As per our guidelines, please add a 'Data Availability Section', where you state that no data were deposited in a public database.
- All articles published beginning 1 July 2020, the EMBO Reports reference style changed to the Harvard style for all article types. Details and examples are provided at <https://www.embopress.org/page/journal/14693178/authorguide#referencesformat>. Please update the reference style accordingly.
- We note that the funding information in the manuscript submission system is incomplete.
- Table 1 is currently called out as Table I. Please change it to Table 1.
- Please remove the black boxes behind scale bars in Figure 1.
- Please provide higher resolution versions of the graphs of Figure EV 1D.
- Please rename the 'Methods' section to 'Materials and Methods'.
- Papers published in EMBO Reports include a 'Synopsis' to further enhance discoverability. Synopses are displayed on the html version of the paper and are freely accessible to all readers. The synopsis includes a short standfirst summarizing the study in 1 or 2 sentences that summarize the key findings of the paper and are provided by the authors and streamlined by the handling editor. I would therefore ask you to include your synopsis blurb.
- In addition, please provide an image for the synopsis. This image should provide a rapid overview of the question addressed in the study but still needs to be kept fairly modest since the image size cannot exceed 550x400 pixels.
- Our production/data editors have asked you to clarify several points in the figure legends (see attached document). Please incorporate these changes in the attached word document and return it with track changes activated.

Thank you again for giving us to consider your manuscript for EMBO Reports, I look forward to your minor revision.

Kind regards,

Deniz

--

Deniz Senyilmaz Tiebe, PhD
Editor
EMBO Reports

Referee #1:

Veliova et al. have thoroughly addressed my concerns from the first round of review. The new experiments have clearly improved the MS, and congratulations to the authors for completing them during the challenges of covid. My only remaining concern is that the revised text refers to mitochondrial glutaminase (Gls2) activity as a necessary component of their model. While Gls2 is abundant in the liver, its expression is low to zero in other tissues. It is important for the authors to consider this in their model and address the issue of minimal Gls2 in BAT. I believe this can be textually. But, for adequacy, if such textual modifications retain mitochondrial glutaminase in the model, they must be made within a quantitative framework and directly address the issue of whether mitochondrial glutaminase is present as high enough levels in BAT to support the model.

Referee #2:

The authors have nicely and carefully addressed all the points I had raised before. The manuscript has been significantly strengthened and is, according to me, suitable for publication.

Referee #1:

Veliova et al. have thoroughly addressed my concerns from the first round of review. The new experiments have clearly improved the MS, and congratulations to the authors for completing them during the challenges of covid. My only remaining concern is that the revised text refers to mitochondrial glutaminase (Gls2) activity as a necessary component of their model. While Gls2 is abundant in the liver, its expression is low to zero in other tissues. It is important for the authors to consider this in their model and address the issue of minimal Gls2 in BAT. I believe this can be textually. But, for adequacy, if such textual modifications retain mitochondrial glutaminase in the model, they must be made within a quantitative framework and directly address the issue of whether mitochondrial glutaminase is present as high enough levels in BAT to support the model.

Answer:

We would like to thank the reviewer for bringing up this important point and giving us the opportunity to clarify. Indeed, we wrote in our revised version that mitochondrial glutaminase could explain partial reversal of the effects of MPC inhibition when Aralar was knocked down, however this should be corrected to glutaminase activity in general (mitochondrial Gls2 or cytosolic Gls1). As the reviewer pointed out mitochondrial Gls2 expression is very low but still detectable in brown adipocytes, while cytosolic Gls1 expression is high in BAT is comparable to heart and liver (10.1074/mcp.M112.024919, 10.1016/j.cmet.2009.08.014). Our data does not definitively point towards either of the two isoforms, but solely suggest that glutaminase activity might be increased upon MPC inhibition. Furthermore, it is conceivable that increased Aralar2 expression might explain why Aralar1 KD only partially reversed the UK5099 effects on aspartate/glutamate ratio. Indeed Aralar1 is the predominant isoform expressed in BAT compared to Aralar 2 under thermoneutral conditions (~5 times higher), which is why we chose to knock down this isoform. However, Aralar2 expression increases ~4 times after 4 days of cold exposure (<https://pubmed.ncbi.nlm.nih.gov/19808025/>). Thus, it is possible that Aralar2 expression increases to compensate for the lack of Aralar1 as our experiments were performed 5-6 days after silencing. However, i) we do not have any experimental evidence to support this and ii) even if that was the case, the affinities of Aralar1 and Aralar2 for glutamate are roughly the same (<https://www.ncbi.nlm.nih.gov/pmc/articles/PMC125626/>). It is thus possible that a combination of low GLS2 activity, low Aralar2 and the Aralar1 left after KD, together with increased GLS1 activity, could account for partial reversal of UK5099 effects after Aralar1 KD. Further studies are needed to identify if there is one factor more important than the others. We thus changed the discussion of Aralar knock down results to: *“Knock-down of Aralar1 resulted in partial reversal of the increase in aspartate:glutamate ratio induced by MPC inhibition (Fig 4I). Partial reversal was expected, given that glutamine can provide glutamate and alpha-ketoglutarate independently of the MASH, through glutaminase and glutamate dehydrogenase.”*

Referee #2:

The authors have nicely and carefully addressed all the points I had raised before. The manuscript has been significantly strengthened and is, according to me, suitable for publication.

Answer: We thank the reviewer for her/his time evaluating the manuscript and the constructive comments. Addressing the points raised by the reviewer made are manuscript stronger and suitable for publication.

Dear Orian,

Thank you for submitting your revised manuscript. I have now looked at everything and all looks fine. Therefore I am very pleased to accept your manuscript for publication in EMBO Reports.

Congratulations on a nice study!

Kind regards,

Deniz

--

Deniz Senyilmaz Tiebe, PhD
Editor
EMBO Reports

--

At the end of this email I include important information about how to proceed. Please ensure that you take the time to read the information and complete and return the necessary forms to allow us to publish your manuscript as quickly as possible.

As part of the EMBO publication's Transparent Editorial Process, EMBO reports publishes online a Review Process File to accompany accepted manuscripts. As you are aware, this File will be published in conjunction with your paper and will include the referee reports, your point-by-point response and all pertinent correspondence relating to the manuscript.

If you do NOT want this File to be published, please inform the editorial office within 2 days, if you have not done so already, otherwise the File will be published by default [contact: emboreports@embo.org]. If you do opt out, the Review Process File link will point to the following statement: "No Review Process File is available with this article, as the authors have chosen not to make the review process public in this case."

Should you be planning a Press Release on your article, please get in contact with emboreports@wiley.com as early as possible, in order to coordinate publication and release dates.

Thank you again for your contribution to EMBO reports and congratulations on a successful publication. Please consider us again in the future for your most exciting work.

THINGS TO DO NOW:

You will receive proofs by e-mail approximately 2-3 weeks after all relevant files have been sent to our Production Office; you should return your corrections within 2 days of receiving the proofs.

Please inform us if there is likely to be any difficulty in reaching you at the above address at that time. Failure to meet our deadlines may result in a delay of publication, or publication without your corrections.

All further communications concerning your paper should quote reference number EMBOR-2019-49634V3 and be addressed to emboreports@wiley.com.

Should you be planning a Press Release on your article, please get in contact with emboreports@wiley.com as early as possible, in order to coordinate publication and release dates.

Corresponding Author Name: Orian S Shirihai, Marcus F. Oliveira, Marc Liesa

Manuscript Number: EMBOR-2019-49634V1